# Edit Less, Achieve More: Dynamic Sparse Neuron Masking for Lifelong Knowledge Editing in LLMs

**Jinzhe Liu**[1,2]   **Junshu Sun**[1,2]   **Shufan Shen**[1,2]   **Chenxue Yang**[3*]   **Shuhui Wang**[1*]

[1]State Key Lab. of AI Safety, Institute of Computing Technology, Chinese Academy of Sciences
[2]University of Chinese Academy of Sciences   [3]Agriculture Information Institute, CAAS
{liujinzhe23b, sunjunshu21s, shenshufan22z, wangshuhui}@ict.ac.cn
yangchenxue@caas.cn

## Abstract

Lifelong knowledge editing enables continuous, precise updates to outdated knowledge in large language models (LLMs) without computationally expensive full retraining. However, existing methods often accumulate errors throughout the editing process, causing a gradual decline in both editing accuracy and generalization. To tackle this problem, we propose **N**euron-Specific **M**asked **K**nowledge **E**diting (**NMKE**), a novel fine-grained editing framework that combines neuron-level attribution with dynamic sparse masking. Leveraging neuron functional attribution, we identify two key types of knowledge neurons, with knowledge-general neurons activating consistently across prompts and knowledge-specific neurons activating to specific prompts. NMKE further introduces an entropy-guided dynamic sparse mask, locating relevant neurons to the target knowledge. This strategy enables precise neuron-level knowledge editing with fewer parameter modifications. Experimental results from thousands of sequential edits demonstrate that NMKE outperforms existing methods in maintaining high editing success rates and preserving model general capabilities in lifelong editing. Codes are provided in https://github.com/LiuJinzhe-Keepgoing/NMKE.

## 1 Introduction

Lifelong model editing has emerged as an effective paradigm for maintaining and iterating modern large language models (LLMs), which enables continual and dynamic knowledge injection, error correction, and sensitive content removal [1, 2, 3, 4]. For example, model editing can rectify outdated knowledge in LLMs without complete retraining, such as updating *the year of the next Olympic Games* from *2024* to *2028*. An ideal lifelong knowledge editing method must simultaneously maintain high editing accuracy while preserving the model's general capabilities [3, 5]. Previous knowledge editing methods fall into two fundamental categories: those that integrate external parameters [5, 6, 7, 8, 9] and those that directly modify internal model parameters [10, 11, 12, 13, 14, 15]. External parameters methods demonstrate strong generalization, but suffer from escalating resource overheads and progressively declining editing accuracy as the number of edits increases [16]. In contrast, internal parameter methods, which typically follow a locate-then-edit paradigm [17], offer enhanced interpretability and a simpler architecture [18], yet are prone to degradation in general capabilities [15, 19]. More critically, both types of methods face fundamental limitations in lifelong editing scenarios, where error accumulation compounds exponentially with successive editing operations [19, 20, 21]. This creates a compelling research challenge: how to design a method that harmonizes the strong generalization of external approaches with the efficiency of internal methods, while enabling robust lifelong editing without performance degradation.

---

*Corresponding author.

39th Conference on Neural Information Processing Systems (NeurIPS 2025).

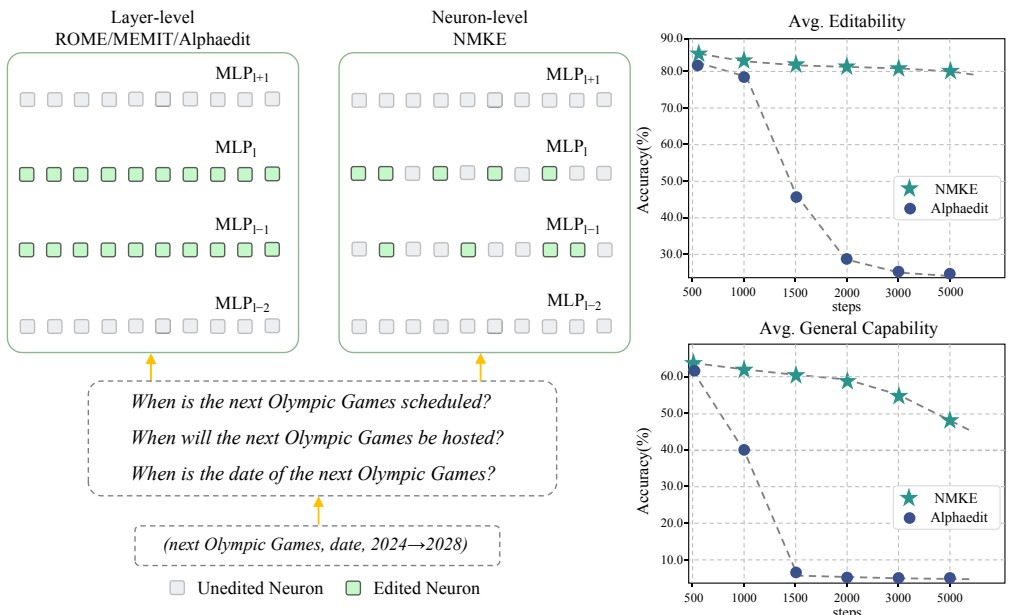

Figure 1: **Left**: Comparison of layer-level editing with our neuron-level editing method, **NMKE**, highlighting the impact on Multilayer Perceptron (MLP) layers. **Right**: Average editability score (including edit success rate [23], generalization success rate and localization success rate [3]) and average general capability (across MMLU [24], GSM8K [25], CommonsenseQA [26], and BBH-Zeroshot [27]) across different edit steps on LLaMA3-8B-Instruct [28].

Motivated by this challenge, we focus on internal editing methods to ensure a simplified architecture, aiming to achieve high editing accuracy while preserving the model's general capabilities. Previous internal methods have explored various localization strategies such as causal tracing [13], multi-layer updates [14], and null-space projection [15] to identify factual knowledge within model parameters. However, these strategies apply modifications at the level of entire layers or parameter blocks, inadvertently affecting neurons unrelated to the target knowledge and consequently causing model forgetting and capability collapsing [15, 22], as shown in Figure 1.

In this paper, we propose **N**euron-specific **M**asked **K**nowledge **E**diting (**NMKE**), a novel fine-grained knowledge editing framework that performs neuron-level attribution and constructs dynamic sparse masks to precisely modify LLMs knowledge. Unlike previous coarse-grained approaches, NMKE employs neuron-level attribution in the feedforward network (FFN) to quantify precisely how individual neurons contribute to knowledge-based predictions. Through this attribution analysis, we discover two distinct types of knowledge-encoding neurons: (i) *Knowledge-General Neurons*, which exhibit stable activation across prompts and encode generalizable information, and (ii) *Knowledge-Specific Neurons*, which are selectively activated to capture semantic variations in specific contexts. Leveraging this insight, NMKE also constructs dynamic sparse masks, selectively focusing on the most relevant subset of neurons associated with the target knowledge. Comprehensive empirical evaluation demonstrates that NMKE achieves an optimal balance in lifelong editing by making minimal-cost parameter modifications, ensuring accurate edits, and preserving general capabilities. Our research makes the following key contributions:

- We experimentally show that the performance degradation in lifelong editing is driven by the cumulative disruption of neurons caused by coarse-grained parameter updates. Building on this analysis, we identify the critical roles of *Knowledge-General* and *Knowledge-Specific* neurons in storing factual knowledge within the model.

- We introduce **N**euron-Specific **M**asked **K**nowledge **E**diting (**NMKE**), a novel fine-grained editing framework that utilizes neuron-level attribution and dynamic sparse masking to precisely target and modify only those neurons most relevant to the updated knowledge, significantly reducing unintended disruption to the model.

- Through rigorous evaluation on thousands of sequential edits,we show that **NMKE** consistently outperforms existing methods in maintaining high editing success rates while simultaneously preserving model capabilities in lifelong editing scenarios.

## 2 Preliminaries

### 2.1 Lifelong Model Editing

Lifelong model editing enables continuous knowledge updates in pretrained language models, maintaining accuracy and correcting errors without the need for full retraining. Formally, let $f_\theta : \mathcal{X} \to \mathcal{Y}$ be a language model with parameters $\theta$. In the sequential editing process, the $t$-th editing step receives an editing request set $S_t = \{(s, r, o \to o^*)\}$, indicating that the fact triple $(s, r, o)$ should be updated to $(s, r, o^*)$. Here, $s$ represents the subject, $r$ the relation, and $o$ the original and target object, respectively. The model is updated via an editing function $E$, such that $\theta^{(t)} = E(S_t, \theta^{(t-1)})$, where the objective that for all $(s, r, o \to o^*) \in S_t$, the edited model satisfies $f_{\theta^{(t)}}(s, r) = o^*$. Simultaneously, for unedited inputs $x \notin \bigcup_t S_t$, the model output should remain as similar as possible to their previous values $f_{\theta^{(t)}}(x) \approx f_{\theta^{(t-1)}}(x)$. The central challenge in lifelong model editing is to achieve high editing accuracy while minimizing interference with unedited knowledge.

### 2.2 Knowledge Storage in Transformers

Knowledge in Transformers [29] is primarily encoded within the FFN, which functions as distributed key-value associative memories [12, 30, 31]. Formally, an FFN layer with input weights $\mathbf{W}^{\text{in}} \in \mathbb{R}^{d_m \times d}$ and output weights $\mathbf{W}^{\text{out}} \in \mathbb{R}^{d \times d_m}$ transforms an input vector $\mathbf{x} \in \mathbb{R}^d$ through the forward propagation: $\mathbf{y} = \mathbf{W}^{\text{out}} \sigma(\mathbf{W}^{\text{in}}\mathbf{x}) + \mathbf{x}$, where $\sigma(\cdot)$ represents the non-linear activation function, and $d_m$ denotes the dimensionality of the hidden layer. Each neuron in the FFN can be considered a key-value unit, where the $i$-th neuron associates with the key $\mathbf{k}^{(i)} = \mathbf{W}^{\text{in}}_{i,:}$ and the value $\mathbf{v}^{(i)} = \mathbf{W}^{\text{out}}_{:,i}$. When the input $\mathbf{x}$ aligns with $\mathbf{k}^{(i)}$, the neuron is strongly activated, and the corresponding value $\mathbf{v}^{(i)}$ contributes significantly to the output. The FFN can be reformulated in a key-value form:

$$\mathbf{y} = \sum_{i=1}^{d_m} \mathbf{s}^{(i)} + \mathbf{x} = \sum_{i=1}^{d_m} \sigma(\mathbf{k}^{(i)}\mathbf{x})\mathbf{v}^{(i)} + \mathbf{x}, \tag{1}$$

where $\mathbf{s}^{(i)}$ represents the contribution of the $i$-th neuron to the output. This decomposition shows that individual neurons collectively encode factual knowledge, but multiple factual associations often share overlapping parameter subsets. Conventional approaches that modify entire parameter blocks or layers during knowledge editing lead to interference with unrelated representations. In lifelong editing, this interference accumulates, degrading model performance and exacerbating distribution shift. Our key insight is that to maintain editing stability and preserve general capabilities, parameter updates should target neural subspaces directly relevant to the edited knowledge, rather than applying coarse updates to entire layers or modules.

## 3 Neuron-specific Masked Knowledge Editing

This section introduces **NMKE**, a fine-grained framework for knowledge editing that leverages neuron-level attribution and constructs dynamic sparse masks. § 3.1 introduces the neuron-level attribution method, where the neuron activations and masking behavior reveal the functional differences among knowledge neurons and identify the knowledge-general and knowledge-specific neurons. § 3.2 introduces an entropy-based dynamic masking strategy to select minimal intervention subsets.

### 3.1 Neuron Attribution

**Neuron-level Attribution.** Inspired by [32], we estimate neuron importance using a static attribution method. The contribution of each neuron to the predicted token is quantified by the increase in log probability when its activation is perturbed. Specifically, $\mathbf{s}^{(i)}$ in Eq. 1 can be regarded as the perturbation from the $i$-th neuron to the input $\mathbf{x}$. We formulate the perturbing process as:

$$\mathbf{x}^{(i)} = \mathbf{x} + \lambda \cdot \mathbf{s}^{(i)}, \tag{2}$$

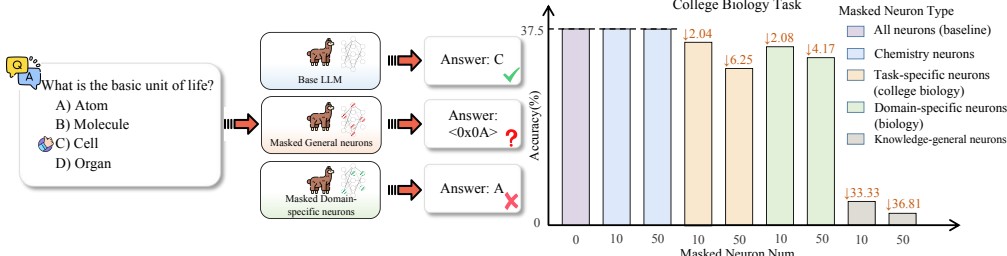

Figure 2: **Left**: The effect of neuron masking on multiple-choice task performance. **Right**: Performance changes following the masking of different neuron types on the MMLU sub-task, with general neuron masking leading to a substantial performance decline, particularly in college biology accuracy, whereas domain-specific neurons cause minimal degradation. ↓ represents the accuracy drop, with the magnitude shown as a percentage.

where $\lambda$ is the amplification factor to scale the perturbation. We then project both $\mathbf{x}$ and $\mathbf{x}^{(i)}$ into the vocabulary space using the unembedding matrix $\mathbf{E}_u$ and then compute the softmax logits, giving the importance score of neuron $i$ as the log-probability gain of the target token $y$:

$$\text{Imp}^{(i)} = \log p(y \mid \mathbf{x}^{(i)}) - \log p(y \mid \mathbf{x}) = \texttt{softmax}(\mathbf{E}_u\mathbf{x}^{(i)}) - \texttt{softmax}(\mathbf{E}_u\mathbf{x}). \tag{3}$$

In practice, the importance scores are computed in batches for each editing prompt, yielding a neuron importance matrix $\mathbf{I} \in \mathbb{R}^{n \times d}$ with $n$ denoting the batch size. This matrix is used to build sparse masks (§ 3.2) that select key neurons for editing, enabling precise and low-interference model updates.

**Functional Roles of Neurons.** To explore the reasons behind capability degradation in lifelong knowledge editing, we conduct attribution experiments on the LLaMA2-7B [33] model. Specifically, we analyze neuron activations in the FFN layers of Transformer blocks across tasks from the MMLU dataset [24], focusing on college and high school level biology and chemistry. Based on cross-task activation patterns, FFN neurons can be categorized into three types: knowledge-general neurons activated across all tasks, domain-specific neurons activated within a subject, and task-specific neurons activated only in one task.

We observe that knowledge-general neurons occur most frequently, followed by domain-specific neurons, with task-specific neurons being the least common. To assess the functional roles of different neuron types, we conduct masking experiments by ablating the top-10 and top-50 neurons with the highest attribution scores for each category and evaluating performance changes on MMLU subtasks. As shown in Figure 2, masking knowledge-general neurons causes severe degradation where the model produces meaningless outputs (*e.g.*, brackets, stop words, or corrupted symbols), with accuracy degrading from 37.5% to 4.17% with a drop of ↓ 33.33% in the top-10 ablation and to 0.69% with ↓ 36.81% drop in the top-50 ablation. In contrast, masking task-specific neurons that correspond to the same domain leads to only minor drops (↓ 2.04% and ↓ 6.25%), while masking domain-specific neurons that correspond to a different domain has a negligible effect. These results suggest that knowledge in LLMs is not uniformly distributed but sparsely localized within a small set of neurons. More experimental details are provided in Appendix B.1.

Building on these observations, we abstract the functional roles of neurons into two categories to guide knowledge editing, *i.e.*, knowledge-general and knowledge-specific neurons. Specifically, neurons with stable activations across semantically similar prompts are defined as knowledge-general neurons, while those with sharp, localized activations and high attribution scores in specific prompts are termed as knowledge-specific neurons. Knowledge-specific neurons include domain-specific and task-specific neurons. Model editing in each step should target the subset of the neurons, excluding irrelevant knowledge-specific neurons.

## 3.2 Dynamic Sparse Masking

To locate relevant neurons for knowledge editing, we introduce a dynamic sparse masking mechanism that adaptively selects a subset of neurons that provide predominant contributions in representing target knowledge. Specifically, let $\mathbf{I}^{(l)} \in \mathbb{R}^{n \times d_l}$ be the neuron attribution matrix at layer $l$, where $n$ is the number of prompts and $d_l$ the number of neurons. The main difference between the two types

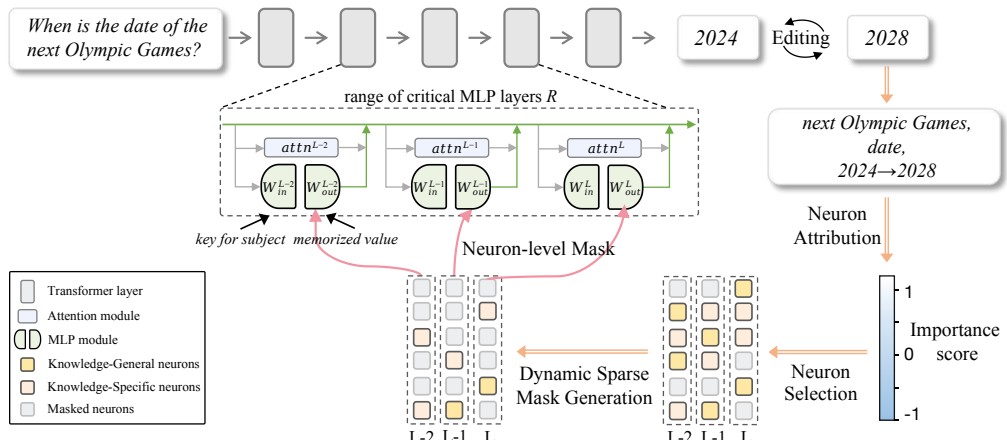

Figure 3: **Overview of NMKE.** Neuron attribution and dynamic sparse masking to selectively update **Knowledge-General** and **Knowledge-Specific** neurons, ensuring precise knowledge editing while preserving the model's general capabilities.

of neurons is that knowledge-general neurons demonstrate consistent activations across different prompts, while knowledge-specific neurons demonstrate selective activations to specific prompts. By analyzing the distribution of attribution scores, we can distinguish between the two types of neurons.

### 3.2.1 Type-specific Selection Score

**Knowledge-general Neurons**, due to their encoding of core knowledge, exhibit stable activations across diverse prompts, resulting in consistently positive attribution scores. In consequence, such neurons can be identified by counting the number of positive attribution scores over multiple prompts:

$$\mathbf{r}_i^{ge} = \sum_{j=1}^{n} \mathbb{I}[\mathbf{I}_{j,i}^{(l)} > 0], \tag{4}$$

where $\mathbb{I}[\cdot]$ denotes the sign function. $\mathbb{I}[\texttt{True}] = 1$ and $\mathbb{I}[\texttt{False}] = 0$.

**Knowledge-specific Neurons**, due to their encoding of task-specific knowledge, are activated on at least one prompt, resulting in a significant attribution score on a specific prompt. Therefore, such neurons can be identified by the maximum attribution across prompts:

$$\mathbf{r}_i^{sp} = \max_{j} \mathbf{I}_{j,i}^{(l)}. \tag{5}$$

### 3.2.2 Type-specific Selection Ratio

To identify the subset of the knowledge-general and knowledge-specific neurons, we select a ratio of the neurons based on their scores $\mathbf{r}^{ge}$ and $\mathbf{r}^{sp}$. However, given different batches of prompts, the neurons perform different activation statuses, and thus resulting in different attribution scores. Employing a fixed ratio for all prompt batches can incorporate irrelevant neurons when fewer neurons are well-activated, and lose relevant neurons when most neurons are well-activated. To tackle this problem, we propose to dynamically assign the selection ratio based on the distribution of the attribution scores.

**Knowledge-general Neurons** demonstrate stable activations across different prompts, indicating a uniform distribution with higher entropy. Therefore, we can employ the average normalized entropy across prompts to evaluate the ratio of the knowledge-general neurons:

$$\mathcal{H}_{\text{ge}} = -\frac{1}{n \log d_l} \sum_{j=1}^{n} \sum_{i=1}^{d_l} \mathbf{P}_{j,i} \log \mathbf{P}_{j,i}, \quad \mathbf{P}_{j,i} = \texttt{softmax}(\alpha \mathbf{I}_{j,i}^{(l)}), \tag{6}$$

where the temperature $\alpha$ adjusts the score differences. The higher the average entropy $\mathcal{H}_{\text{ge}}$, the more neurons can be categorized as knowledge-generalized.

**Knowledge-specific Neurons** demonstrate selective activation, indicating a large maximum attribution score across prompts. When most neurons are categorized as knowledge-specific, the maximum attribution scores will be similarly large across different neurons and demonstrate a large distribution entropy:

$$\mathcal{H}_{\text{sp}} = -\frac{1}{\log d_l} \sum_{i=1}^{d_l} \mathbf{q}_i \log \mathbf{q}_i, \quad \mathbf{q}_i = \frac{\max_j \mathbf{I}_{j,i}^{(l)}}{\sum_{i'} \max_j \mathbf{I}_{j,i'}^{(l)}}. \tag{7}$$

These entropy values indicate the ratio of the relevant neurons in different prompt batches. In practice, we further adjust the ratio by constant scalers $a_{\text{ge}}$, $a_{\text{sp}}$ and bias terms $b_{\text{ge}}$, $b_{\text{sp}}$, giving $\rho_{\text{ge}} = \mathcal{H}_{\text{ge}} \cdot a_{\text{ge}} + b_{\text{ge}}$ and $\rho_{\text{sp}} = \mathcal{H}_{\text{sp}} \cdot a_{\text{sp}} + b_{\text{sp}}$.

**Neuron Selection and Mask Generation.** Based on the estimated ratio $\rho_{\text{ge}}$, we compute the threshold $\tau_{\text{ge}}$ as the $(1 - \rho_{\text{ge}})$ quantile of the score $\mathbf{r}^{\text{ge}}$, and select neurons exceeding this threshold as the knowledge-general neurons. The selection of the knowledge-specific neurons follows the same protocol with ratio $\rho_{\text{sp}}$ and score $\mathbf{r}^{\text{sp}}$. This selection constructs the binary sparse mask $\mathbf{m}^{(l)} \in \{0, 1\}^{d_1}$ as:

$$\mathbf{m}_i^{(l)} = \mathbb{I}[\mathbf{r}_i^{\text{ge}} \geq \tau_{\text{ge}} \quad \text{or} \quad \mathbf{r}_i^{\text{sp}} \geq \tau_{\text{sp}}]. \tag{8}$$

This hybrid selection mechanism provides flexible control of the neuron-level editing by incorporating the functional roles of different neurons. By selectively updating neurons relevant to the edited knowledge, this approach enables precise editing while preserving the knowledge structure of irrelevant neurons. As shown in Figure 3, we propose a neuron-level knowledge editing framework, NMKE, that restricts updates to a subset of functionally important MLP neurons, enabling precise and minimal-disruptive knowledge injection. We follow AlphaEdit [15] to construct the optimization pipeline, where $\mathbf{m}$ is employed to mask the parameter update matrix.

## 4 Experiments

### 4.1 Experimental Setup

**Evaluation Benchmarks.** We evaluate our methods in terms of editing performance and the generalization ability of edited LLMs. For the knowledge editing performance, we utilize two standard benchmarks: ZsRE [34] for question answering, and CounterFact [13] for factual corrections. Following [6, 4], we report three key metrics: Rel. [23] (edit success), Gen. [3] (generalization to paraphrased prompts), and Loc. [3] (locality preservation). For the generalization ability, we adopt five downstream tasks that span mathematical reasoning, question answering, and code generation: MMLU [24], GSM8K [25], CommonsenseQA [26], BBH-Zeroshot [27], and HumanEval [35].

**Baselines.** We compare NMKE with various of baselines that covers both external and internal parameter methods, including Fine-Tuning (FT) [36], KN [12], ROME [13], PMET [37], MEMIT [14], WISE [6], and AlphaEdit [15]. Evaluation is conducted on widely used LLMs including LLaMA3-8B-Instruct [28], GPT2-XL [38], and Qwen2.5-7B [39] across cumulative edit steps $T \in \{10, 100, 500, 1000, 1500, 2000, 3000, 5000\}$. More details are presented in Appendix A.

### 4.2 Does the Edited LLM Still Generalize?

Figure 4 evaluates the generalization ability of LLaMA-3-8B-Instruct on the ZsRE and CounterFact datasets edited by various methods with 1 to 2000 editing steps. Overall, the results show that *existing methods exhibit rapid degradation in generalization ability as the number of edits increases*. FT destroys the generalization ability of LLMs on GSM8K and HumanEval after only 100 edits, while methods like ROME and MEMIT show significant performance drops beyond $T = 500$. As the SOTA method, AlphaEdit suffers from a gradual decline in generalization due to the continuous modification of irrelevant knowledge neurons during layer-level edits. For example, at $T = 1500$, AlphaEdit shows a significant decline in question answering performance, with nearly a complete loss of mathematical and coding abilities, as the accuracy on both GSM8K and HumanEval drops to 0. In contrast, our NMKE effectively maintains the generalization of LLMs in lifelong editing on both datasets and significantly outperforms all baselines with $T > 1000$. The superior performance of NMKE can be attributed to its dynamic sparse masking mechanism, which confines the edited

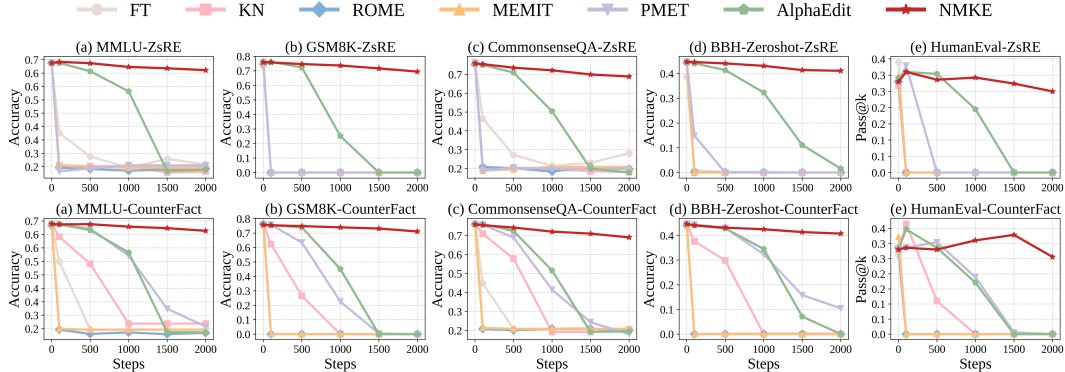

Figure 4: Generalization performance of LLaMA-3-8B-Instruct on MMLU [24], GSM8K [25], CommonsenseQA [26], BBH-Zeroshot [27], and HumanEval [35] benchmarks after 2000 steps of sequential editing on the ZsRE (*top*) and CounterFact (*bottom*) datasets.

Table 1: Main editing results (LLaMA3-8B-Instruct) across 2000 editing steps on ZsRE and Counter-Fact datasets. The best result is in **bold**, and the second-best result is underlined.

| | | | | | | | | | | ZsRE | | | | | | | | | |
|---|---|---|---|---|---|---|---|---|---|---|---|---|---|---|---|---|---|---|
| **Method** | $T=10$ | | | $T=100$ | | | $T=500$ | | | $T=1000$ | | | $T=1500$ | | | $T=2000$ | | |
| | Rel. | Gen. | Loc. | Rel. | Gen. | Loc. | Rel. | Gen. | Loc. | Rel. | Gen. | Loc. | Rel. | Gen. | Loc. | Rel. | Gen. | Loc. |
| FT | 0.18 | 0.03 | 0.01 | 0.17 | 0.13 | 0.03 | 0.12 | 0.11 | 0.00 | 0.13 | 0.10 | 0.02 | 0.12 | 0.10 | 0.02 | 0.07 | 0.06 | 0.01 |
| KN | 0.14 | 0.12 | 0.66 | 0.02 | 0.01 | 0.00 | 0.00 | 0.00 | 0.00 | 0.00 | 0.00 | 0.00 | 0.00 | 0.00 | 0.00 | 0.00 | 0.00 | 0.00 |
| ROME | 0.96 | 0.94 | 0.64 | 0.10 | 0.09 | 0.03 | 0.01 | 0.01 | 0.02 | 0.02 | 0.01 | 0.02 | 0.04 | 0.04 | 0.02 | 0.01 | 0.01 | 0.02 |
| MEMIT | 0.98 | **0.98** | 0.89 | 0.06 | 0.03 | 0.01 | 0.04 | 0.04 | 0.03 | 0.04 | 0.04 | 0.03 | 0.04 | 0.03 | 0.03 | 0.03 | 0.04 | 0.03 |
| PMET | 0.39 | 0.39 | 0.91 | 0.02 | 0.02 | 0.05 | 0.00 | 0.00 | 0.00 | 0.00 | 0.00 | 0.00 | 0.00 | 0.00 | 0.00 | 0.00 | 0.00 | 0.00 |
| WISE | 0.83 | 0.78 | - | 0.71 | 0.67 | - | 0.46 | 0.45 | - | 0.41 | 0.39 | - | 0.32 | 0.31 | - | 0.37 | 0.36 | - |
| AlphaEdit | **0.99** | **0.98** | **0.97** | **0.99** | **0.95** | **0.86** | **0.96** | **0.87** | 0.71 | 0.93 | 0.84 | 0.58 | 0.62 | 0.54 | 0.14 | 0.32 | 0.28 | 0.06 |
| NMKE (Ours) | 0.93 | 0.90 | 0.95 | 0.95 | **0.95** | **0.86** | **0.96** | **0.87** | **0.82** | **0.95** | **0.85** | **0.77** | **0.94** | **0.86** | **0.74** | **0.94** | **0.85** | **0.71** |

| | | | | | | | | | | CounterFact | | | | | | | | | |
|---|---|---|---|---|---|---|---|---|---|---|---|---|---|---|---|---|---|---|
| **Method** | $T=10$ | | | $T=100$ | | | $T=500$ | | | $T=1000$ | | | $T=1500$ | | | $T=2000$ | | |
| | Rel. | Gen. | Loc. | Rel. | Gen. | Loc. | Rel. | Gen. | Loc. | Rel. | Gen. | Loc. | Rel. | Gen. | Loc. | Rel. | Gen. | Loc. |
| FT | 0.02 | 0.00 | 0.00 | 0.04 | 0.00 | 0.00 | 0.07 | 0.01 | 0.00 | 0.03 | 0.01 | 0.00 | 0.05 | 0.01 | 0.00 | 0.07 | 0.03 | 0.00 |
| KN | 0.00 | 0.01 | 0.87 | 0.03 | 0.03 | 0.79 | 0.01 | 0.01 | 0.66 | 0.00 | 0.00 | 0.00 | 0.00 | 0.00 | 0.00 | 0.00 | 0.00 | 0.00 |
| ROME | 0.90 | **0.70** | 0.60 | 0.23 | 0.13 | 0.01 | 0.01 | 0.01 | 0.00 | 0.01 | 0.00 | 0.00 | 0.01 | 0.00 | 0.00 | 0.00 | 0.00 | 0.00 |
| MEMIT | 0.98 | **0.70** | 0.76 | 0.08 | 0.03 | 0.06 | 0.00 | 0.00 | 0.01 | 0.00 | 0.00 | 0.01 | 0.00 | 0.00 | 0.01 | 0.00 | 0.00 | 0.00 |
| PMET | 0.10 | 0.20 | **0.98** | 0.10 | 0.05 | **0.91** | 0.14 | 0.06 | **0.76** | 0.06 | 0.02 | **0.60** | 0.02 | 0.01 | 0.42 | 0.02 | 0.00 | 0.35 |
| AlphaEdit | **1.00** | 0.60 | 0.85 | **0.99** | **0.80** | 0.71 | **0.99** | **0.75** | 0.44 | **0.99** | **0.76** | 0.32 | 0.71 | 0.55 | 0.19 | 0.22 | 0.13 | 0.04 |
| NMKE (Ours) | 0.92 | 0.51 | 0.86 | 0.97 | 0.56 | 0.80 | **0.99** | 0.58 | 0.60 | **0.99** | 0.65 | 0.50 | **0.98** | **0.65** | **0.43** | **0.98** | **0.67** | **0.38** |

parameters and minimizes disruption to the model's internal representations. Notably, after 5000 editing steps, MMLU remains at 0.59, with further results in Appendix B.2.

### 4.3 Do Sparser Neuron Modifications Lead to Better Edits?

Table 1 presents the results of lifelong knowledge editing on 2000 randomly sampled ZsRE and CounterFact examples, edited sequentially with a batch size of 1, using LLaMA3-8B-Instruct. Most methods (*e.g.*, FT, KN, ROME, and MEMIT) exhibit significant performance degradation after $T = 100$ in lifelong editing. WISE adds external modules without modifying internal parameters, with locality indicated by a "-", but its knowledge editing success rate decreases by ↓ 0.46 after 2000 edits. AlphaEdit's performance becomes unstable during sequential editing, with catastrophic forgetting observed at $T > 1500$, as evidenced by a decline in editing success accuracy of ↓ 0.67 on ZsRE and ↓ 0.78 on CounterFact. In contrast, NMKE demonstrates superior lifelong knowledge editing capabilities on both datasets. This robustness stems from our fine-grained, neuron-level editing strategy, which precisely targets a relevant neuron subset, preventing model collapse. Results for scaling to 5000 are shown in Table 6, with additional experiments in Appendices B.2, B.4 and B.9.

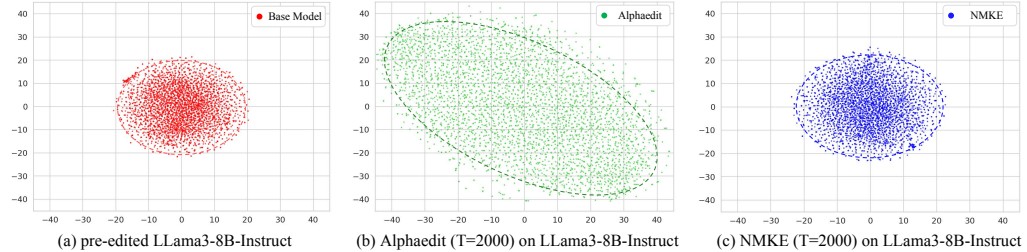

(a) pre-edited LLama3-8B-Instruct    (b) Alphaedit (T=2000) on LLama3-8B-Instruct    (c) NMKE (T=2000) on LLama3-8B-Instruct

Figure 5: Visualization of the distributional shift of down-projection weights in the 8th MLP layer for (a) the pre-edited LLaMA-3-8B-Instruct, (b) AlphaEdit, and (c) NMKE after 2000 sequential edits.

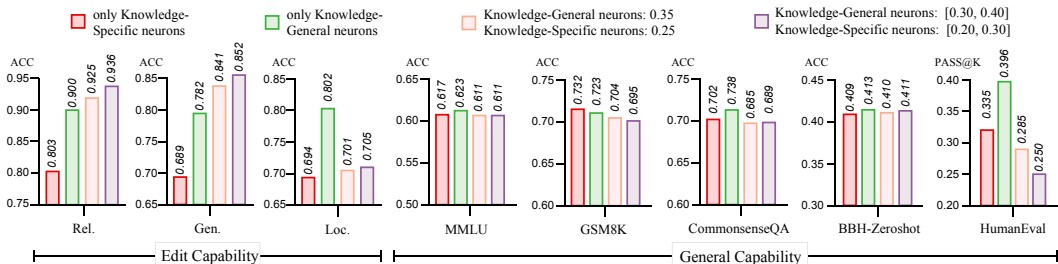

Figure 6: Editing and generalization performance after $T=2000$ sequential edits under various masking strategies.

## 4.4 What Happens to LLM Internal Parameters?

To analyze the impact of different methods on the model's internal parameters, we visualize the distributional shift of down-projection weights in the 8-th FFN layer using t-SNE [40]. We compare the pre-edited LLaMA3-8B-Instruct model with its versions edited by AlphaEdit and NMKE, each after 2000 sequential edits on ZsRE. As shown in Figure 5, the pre-edited LLaMA3-8B-Instruct model (Figure 5 (a)) exhibits a compact weight distribution, indicating a stable parameter structure. AlphaEdit (Figure 5 (b)) induces a noticeable deviation from the original weight distribution, resulting in a more dispersed and distorted geometry in the t-SNE space, suggesting substantial distributional shifts. In comparison, NMKE (Figure 5 (c)) maintains a more compact structure, closely aligned with the original distribution, reflecting minimal parameter shifts. These results suggest that NMKE induces fewer disruptions to the model's internal parameters than AlphaEdit, which explains its superior editing stability and reduced interference, as observed in § 4.2 and § 4.3. Additional analyses of LLM internal parameters are presented in Appendices B.3, B.5, and B.8.

## 4.5 Further Analysis

**Effects of Neuron Selection Strategies.**    Figure 6 compares four neuron selection strategies: activating only *(i)* knowledge-general or *(ii)* knowledge-specific neurons, and activating both types with *(iii)* fixed ratio or *(iv)* entropy-based dynamic ratio. Results show that the entropy-based dynamic ratio achieves the highest editing success rate and best generalization, benefiting from its dynamic neuron activation based on different knowledge types. Notably, activating only knowledge-general neurons excels in locality preservation and retention of coding abilities. Overall, the four neuron selection strategies within the NMKE framework effectively preserve strong general capabilities after 2000 editing steps. Among these, the entropy-based dynamic ratio activation achieves the best balance between enhancing knowledge editing performance and maintaining general capabilities.

**Hyperparameter Analysis for NMKE.**    As shown in Figure 7 (a), in order to investigate the impact of constant scalers $a_{ge}$, $a_{sp}$ and bias terms $b_{ge}$, $b_{sp}$ on the proportion of neuron activation, we adjust the constant scalers $a_{ge} \in \{0.1, 0.5, 1.0\}$ and bias terms $b_{ge} \in \{0.2, 0.3, 0.4\}$ to explore their effects on knowledge editing performance. The results show that higher bias terms enhance the selection of knowledge-general neurons, crucial for stable, generalizable knowledge, leading to more precise updates. In contrast, larger constant scalers broaden neuron activation, which may select unnecessary neurons and increase interference. This approach effectively targets knowledge neurons,

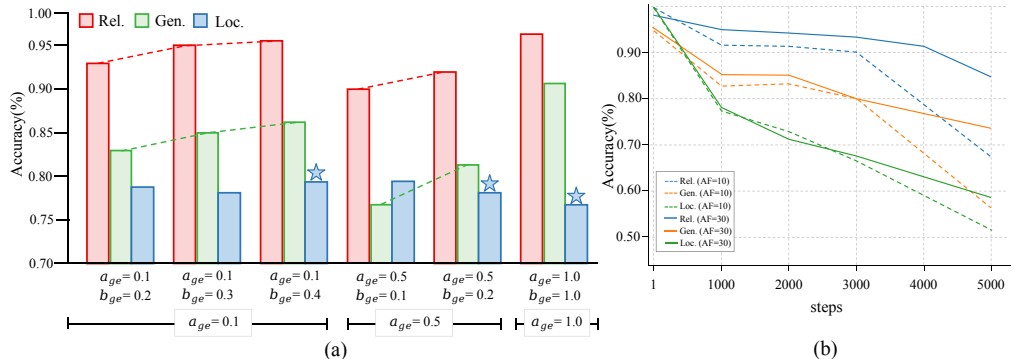

Figure 7: (a) Effect of constant scalers and bias terms on neuron selection. The red and green lines show that higher bias terms improve editing accuracy, while ★ indicates that larger constant scalers reduce locality preservation. (b) Sensitivity of neuron editing to amplification factor $\lambda$.

Table 2: Per-edit runtime and editing performance in sequential editing.

| Method | Step Time (s) | $T = 1000$ | | | $T = 2000$ | | |
|---|---|---|---|---|---|---|---|
| | per edit | Rel. | Gen. | Loc. | Rel. | Gen. | Loc. |
| MEMIT | 16.83 | 0.04 | 0.04 | 0.03 | 0.04 | 0.03 | 0.03 |
| WISE | 26.49 | 0.41 | 0.39 | - | 0.32 | 0.31 | - |
| AlphaEdit | 22.16 | 0.93 | 0.84 | 0.58 | 0.62 | 0.54 | 0.14 |
| NMKE (MPC) | 22.25 | 0.94 | 0.82 | **0.81** | 0.93 | 0.83 | **0.77** |
| NMKE (PSA) | 29.67 | **0.95** | 0.84 | 0.79 | **0.94** | **0.86** | 0.71 |
| NMKE (LPS) | 30.42 | **0.95** | **0.85** | 0.77 | **0.94** | **0.86** | 0.74 |

ensuring precise selection and preserving the model's capabilities while minimizing disruption. In Figure 7 (b), we analyzed the impact of amplification factors $\lambda = 10$ and $\lambda = 30$ on knowledge editing performance. The results show that a moderate increase in the amplification factor helps prioritize neurons most relevant to the target knowledge, improving editing efficiency and reducing disturbances. Additional experimental results are presented in Appendices B.1 and B.6.

**Editing Efficiency.** To quantify attribution cost, we report per-edit runtime under sequential editing across several competitive baselines and NMKE configured with three attribution strategies: MLP Projection Coefficient (MPC) [30], Probability Shift Attribution (PSA) [32], and Log Probability Shift (LPS) [32]. Table 2 shows that NMKE (MPC) is the most efficient variant, with only $\sim 0.2\,\mathrm{s}$ of per-edit overhead, while the highest-performing variant, NMKE (LPS), adds $\sim 8\,\mathrm{s}$ per edit but remains practical and delivers significant improvements. Overall, NMKE is a flexible framework that supports attribution methods of varying complexity, balancing quality and efficiency under different scenarios. More analysis are provided in Appendix B.7.

**Effect of Overlapping Neurons.** To investigate the effect of overlapping neurons, we perform edits using only overlapping neurons or only non-overlapping neurons. Table 3 shows that overlapping neurons mediate the trade-off between edit success and locality. Editing with overlapping neurons yields the strongest locality, with moderate drops in reliability and generalization. Editing with non-overlapping neurons increases edit success and generalization but weakens locality. NMKE combines the two via an entropy-guided ratio and achieves the best overall balance at $T = 1000$ and $T = 2000$, supporting its stability under sequential editing. Further experiments on knowledge neuron distributions are reported in Appendix B.6.

**Neuron Masking Analysis.** To assess the effectiveness of a discretized neuron update mechanism, we compare binary masking with a soft mask alternative. The NMKE (soft-mask) variant scales each neuron's update magnitude by its attribution score, preserving the continuity of neuron states. As shown in Table 4, the soft mask largely matches NMKE in accuracy and generalization, but its dense and indiscriminate updates degrade locality. In contrast, NMKE constrains updates to a sparse, knowledge-aware neuron subset, thereby preserving stability and achieving robust locality.

Table 3: Functional analysis of overlapping neurons in sequential editing.

| Method | $T = 1000$ | | | $T = 2000$ | | |
|---|---|---|---|---|---|---|
| | Rel. | Gen. | Loc. | Rel. | Gen. | Loc. |
| Overlapping Neurons Only | 0.72 | 0.60 | **0.84** | 0.75 | 0.63 | **0.80** |
| Non-Overlapping Neurons Only | 0.82 | 0.70 | 0.82 | 0.85 | 0.74 | 0.78 |
| NMKE | **0.95** | **0.85** | 0.77 | **0.94** | **0.86** | 0.74 |

Table 4: Effectiveness analysis of neuron masking mechanisms.

| Method | $T = 1000$ | | | $T = 2000$ | | |
|---|---|---|---|---|---|---|
| | Rel. | Gen. | Loc. | Rel. | Gen. | Loc. |
| NMKE (soft-mask) | **0.96** | **0.87** | 0.67 | 0.72 | 0.61 | 0.19 |
| NMKE | 0.95 | 0.85 | **0.77** | **0.94** | **0.86** | **0.74** |

## 5 Related Work

**External Parameter Editing.** External parameter editing methods are widely adopted for their structural flexibility and ease of implementation [4, 3, 41, 42]. Early approaches like SERAC [7] employ cached counterfactual models with scope classifiers for relevance-based routing. To improve flexibility, GRACE [5] introduces discrete key-value adaptors for lifelong editing, while Melo [9] proposes semantic-clustered low-rank adaptation modules. ATBias [43] shifts towards in-context editing by biasing key entity tokens during decoding, and WISE [6] enhances the paradigm with dual-memory architectures and trainable routers to isolate edited knowledge. Despite these advances, these methods overlook dependencies among factual updates. To address this, recent methods [44, 45] incorporate knowledge graphs with graph neural networks [46, 47, 48, 49, 50, 51] to capture dependencies between related edits. Nevertheless, external methods that rely on auxiliary modules still suffer from growing storage and routing bottlenecks with more edits, degrading accuracy and efficiency.

**Internal Parameter Editing.** Internal parameter editing approaches directly modify model weights through constrained updates. Early meta-learning methods [11, 10] leverage hypernetworks to generate task-specific parameter updates, but face limitations in scalability when handling sequential edits due to extensive retraining requirements. F-Learning [52] adopts a two-stage forgetting-before-learning fine-tuning paradigm for knowledge updating. The locate-then-edit paradigm [17] has significantly advanced internal editing, evolving from single-layer approaches like ROME [13], which targets specific key-value pairs in transformer layers, to multi-layer methods like MEMIT [14], which distributes updates across multiple layers for enhanced robustness. Furthermore, AlphaEdit [15] employs null-space projection to reduce distribution shift. These approaches operate at the layer or parameter-block level, risking disruption of unrelated neurons and causing catastrophic forgetting and performance degradation in lifelong editing [2, 19]. FiNE [53] performs fine-grained editing via contribution scoring, without explicitly modeling the functional roles of knowledge neurons. Motivated by this, we propose NMKE, which performs fine-grained editing of knowledge neurons via sparse masking, achieving stable lifelong editing while preserving generalization.

## 6 Conclusion

This paper introduces **N**euron-specific **M**asked **K**nowledge **E**diting (**NMKE**), a fine-grained framework for knowledge editing that leverages neuron-level attribution and dynamic sparse masking to enable precise editing. By identifying and targeting both knowledge-general and knowledge-specific neurons, NMKE confines updates to relevant neural subsets, effectively minimizing interference. Experiments demonstrate that NMKE outperforms existing methods in edit success and generalization retention, particularly in lifelong editing scenarios involving thousands of edits, where other methods experience cumulative degradation. This superior performance arises from our neuron-level intervention approach, which ensures successful edits while preserving general capabilities, presenting a promising solution for lifelong editing. For limitations and discussion, please refer to Appendix C.

## Acknowledgments and Disclosure of Funding

This work was supported in part by the National Key R&D Program of China under Grant 2023YFC2508704, and in part by the National Natural Science Foundation of China 62236008. The authors would like to thank the anonymous reviewers for their constructive comments and suggestions that improved this manuscript.

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

# Appendix

In the Appendix, we provide further details, including experimental setups, additional results, analyses, and discussions, as outlined below:

- Appendix A: Experimental setups (cf. Section 4).
- Appendix B: More experimental results (cf. Section 3 and 4).
- Appendix C: Limitations and future discussions.

## A Implementation Details

### A.1 Description of Datasets

We evaluate general model capabilities using five challenging benchmarks covering factual knowledge, reasoning, and code generation. We also test knowledge editing using two standard factual-editing datasets. The benchmarks are described below:

**MMLU** [24]: a multiple-choice benchmark with over 16,000 questions across 57 academic and professional subjects (e.g., mathematics, history, law, medicine). Each question has four answer choices. MMLU tests a model's recall of domain-specific knowledge and multi-domain reasoning, serving as a broad measure of general knowledge and reasoning ability.

**GSM8K** [25]: contains roughly 8,500 grade-school math word problems requiring step-by-step arithmetic reasoning. This benchmark tests a model's ability to perform multi-step numerical calculations.

**CommonsenseQA** [26]: a multiple-choice question-answering dataset focused on everyday commonsense reasoning. Each question has five options, only one of which is correct. CommonsenseQA assesses a model's understanding of implicit context and everyday knowledge.

**BBH** [27]: a subset of 23 challenging tasks drawn from the BIG-Bench benchmark, including logic puzzles, mathematical reasoning, and code comprehension. We evaluate BBH in a zero-shot setting to test a model's general reasoning ability and robustness on complex tasks without any task-specific training. In this paper, we choose to conduct the evaluation using a zero-shot approach.

**HumanEval** [35]: a code-generation benchmark with 164 Python programming problems. Each problem provides a function signature, docstring, and example input-output pairs; the model must generate code that passes the provided unit tests. HumanEval measures a model's functional correctness and programming proficiency.

For factual knowledge editing, we adopt two standard benchmarks following prior work [6, 15]. **ZsRE** [34] is a relation-centric question-answering dataset where each example includes an edit prompt with a target answer, a semantically equivalent paraphrase to test generalization, and an unrelated prompt to probe locality. **CounterFact** [13] consists of factual statements paired with counterfactual versions (created by replacing the subject entity while keeping the predicate fixed).

### A.2 Reproduction Details

In this section, we provide detailed information to reproduce our experimental results. All experiments are conducted using 8 NVIDIA A100 GPUs.

- For all of the models, we use HuggingFace Transformers by default (https://github.com/huggingface/transformers).
- For editing language models, we use the EasyEdit framework (https://github.com/zjunlp/EasyEdit).
- For evaluating the general capabilities of models, we use the Language Model Evaluation Harness (https://github.com/EleutherAI/lm-evaluation-harness).
- In editing experiments, our hyperparameters follow the settings provided by the EasyEdit framework (https://github.com/zjunlp/EasyEdit/tree/main/hparams).

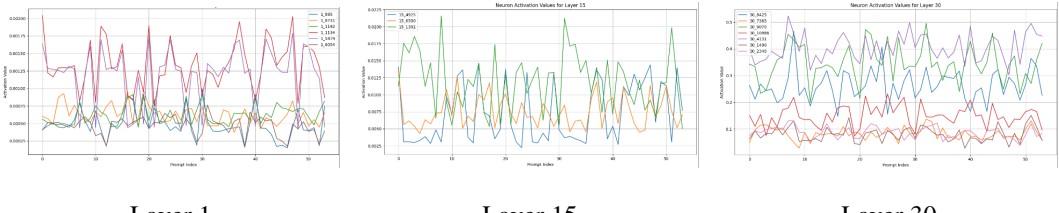

| Layer 1 | Layer 15 | Layer 30 |

Figure 8: Neuron activation values in layers 1, 15, and 30 across 60 prompts.

# B    More Experimental Results and Analyses

## B.1    Functional Roles of Neurons.

To further verify the generality of our neuron classification scheme across domains, we extend the masking analysis from college biology to the high school chemistry subset in MMLU. As shown in Table 5, masking the top-10 and top-50 task-relevant neurons (based on attribution scores) results in accuracy drops of ↓ 14.28% and ↓ 14.77%, respectively. In contrast, masking knowledge-general neurons leads to a drastic degradation, with accuracy dropping by over 35 percentage points. These results are consistent with the findings presented in § 3.1, offering additional empirical evidence for the non-uniform distribution of knowledge neurons within LLMs.

Table 5: Accuracy and drop in performance for different masking strategies on the high school chemistry task.

| Task | Mask Type | Accuracy (%) | Drop (%) |
|------|-----------|--------------|----------|
| high school chemistry | All neurons (base model) | 36.94 | – |
| | high school chemistry (mask top-10) | 22.66 | ↓14.28 |
| | high school chemistry (mask top-50) | 22.17 | ↓14.77 |
| | common neurons (mask top-10) | 1.48 | ↓35.46 |
| | common neurons (mask top-50) | 1.47 | ↓35.47 |

In addition, we further explore the dynamics of neuron activations across different layers. Specifically, we visualize the activation patterns of selected neurons in layers 1, 15, and 30, as shown in Figure 8. Neurons in the early layers exhibit low-amplitude, noisy activations, which reflect their general-purpose characteristics. In contrast, neurons in the mid-to-high layers show higher activation magnitudes and greater variance across prompts, indicating an increased degree of specialization and functional diversity. Notably, several neurons in layer 30 consistently exhibit strong activations across prompts, suggesting a resonance-like behavior and highlighting their potential role in stable knowledge retrieval in specific tasks. These findings further validate our classification of knowledge-general neurons and knowledge-specific neurons, and support the use of dynamic sparse masking to target minimal yet effective subspaces for knowledge editing.

## B.2    Scaling to 5000 Edits: Lifelong Robustness Evaluation.

Table 6 presents a comparison of knowledge editing and generalization performance for AlphaEdit and NMKE on the ZsRE and CounterFact datasets, evaluated at 3000 and 5000 editing steps. On the ZsRE dataset, NMKE consistently outperforms AlphaEdit across all metrics at $T = 3000$. This trend is maintained at $T = 5000$, where NMKE achieves strong performance in both edit success and generalization, while AlphaEdit's performance remains low across both metrics. These results demonstrate NMKE's effectiveness in maintaining generalization and achieving higher accuracy in knowledge editing tasks, even with extensive editing steps.

## B.3    MLP Weight Distribution Changes from 1000 to 2000 Edits: AlphaEdit vs. NMKE

Figure 9 shows the evolution of MLP weight distributions for AlphaEdit and NMKE at $T = 1000$, $T = 1500$, and $T = 2000$. For AlphaEdit, the weight distribution becomes progressively more dispersed with each additional editing step, reflecting the broader, layer-level modifications applied by this method, which leads to increased instability in the model's internal representations. In contrast,

Table 6: Scaling to 5K edits on ZsRE and CounterFact datasets using LLaMA3-8B-Instruct.

| Method | ZSRE (*T*=3000) | | | General Tasks (*T*=3000) | | | | |
|---|---|---|---|---|---|---|---|---|
| | Rel. | Gen. | Loc. | MMLU | GSM8K | CommonsenseQA | BBH-Zeroshot | HumanEval |
| AlphaEdit | 0.17 | 0.14 | 0.02 | 0.26 | 0.00 | 0.21 | 0.00 | 0.00 |
| **NMKE (ours)** | **0.92** | **0.81** | **0.68** | **0.59** | **0.64** | **0.68** | **0.40** | **0.26** |
| Method | ZSRE (*T*=5000) | | | General Tasks (*T*=5000) | | | | |
| | Rel. | Gen. | Loc. | MMLU | GSM8K | CommonsenseQA | BBH-Zeroshot | HumanEval |
| AlphaEdit | 0.02 | 0.02 | 0.00 | 0.27 | 0.00 | 0.20 | 0.00 | 0.00 |
| **NMKE (ours)** | **0.86** | **0.74** | **0.59** | **0.53** | **0.40** | **0.62** | **0.35** | **0.24** |
| Method | CF (*T*=3000) | | | General Tasks (*T*=3000) | | | | |
| | Rel. | Gen. | Loc. | MMLU | GSM8K | CommonsenseQA | BBH-Zeroshot | HumanEval |
| AlphaEdit | 0.04 | 0.01 | 0.00 | 0.23 | 0.00 | 0.20 | 0.00 | 0.00 |
| **NMKE (ours)** | **0.95** | **0.65** | **0.33** | **0.59** | **0.67** | **0.65** | **0.39** | **0.23** |

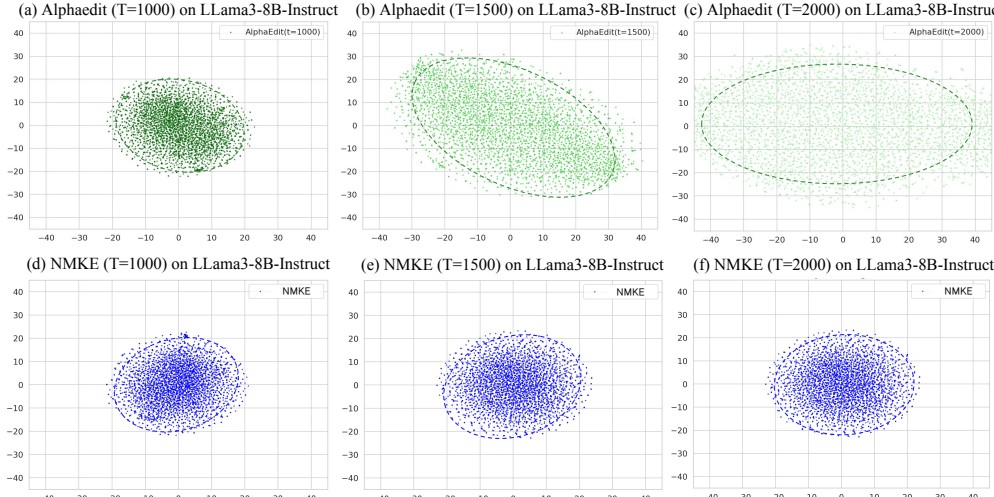

Figure 9: Hierarchical Distribution of Knowledge Neurons Across Layers

NMKE maintains a more stable distribution across all editing steps, with minimal deviation from the original model. This evidences NMKE's neuron-level editing, targeting only relevant neurons and preserving global parameter stability with minimal interference. The observed trends highlight NMKE's advantage in sustaining internal stability across sequential knowledge edits.

## B.4 Editing Performance on GPT2-XL and Qwen2.5-7B

Table 7 shows that NMKE consistently outperforms all baseline methods, including AlphaEdit, on the ZsRE dataset using GPT2-XL, particularly in editing accuracy and localization. While NMKE consistently leads, the difference compared to AlphaEdit is modest due to the smaller size of GPT2-XL. Despite this, NMKE demonstrates a slight advantage, especially in later editing steps. Results on Qwen2.5-7B are shown in Table 8.

## B.5 $\ell_2$-Norm Distribution of Layer 8 MLP Down-Projection Weights

Figure 10 illustrates the $\ell_2$-norm distribution of the layer-8 MLP down-projection weights for the original model, AlphaEdit, and NMKE after 2000 sequential edits. The original model exhibits a tight distribution. In contrast, AlphaEdit broadens and shifts the distribution, indicating large layer-wise perturbations and reduced stability. NMKE remains close to the original with slight deviations, consistent with selective neuron-level editing that minimizes collateral changes. Overall, NMKE induces fewer disruptions to internal representations than AlphaEdit, thereby better preserving stability and general capability under extensive edits.

Table 7: Editing performance on the ZsRE dataset using GPT2-XL across sequential steps.

| | ZsRE on GPT2-XL | | | | | | | | | | | | | | | | | |
|---|---|---|---|---|---|---|---|---|---|---|---|---|---|---|---|---|---|---|
| **Method** | $T=10$ | | | $T=100$ | | | $T=500$ | | | $T=1000$ | | | $T=1500$ | | | $T=2000$ | | |
| | Rel. | Gen. | Loc. | Rel. | Gen. | Loc. | Rel. | Gen. | Loc. | Rel. | Gen. | Loc. | Rel. | Gen. | Loc. | Rel. | Gen. | Loc. |
| FT | 0.26 | 0.18 | 0.06 | 0.06 | 0.06 | 0.02 | 0.07 | 0.04 | 0.01 | 0.06 | 0.05 | 0.01 | 0.06 | 0.06 | 0.01 | 0.05 | 0.04 | 0.00 |
| KN | 0.02 | 0.00 | 0.05 | 0.01 | 0.00 | 0.01 | 0.00 | 0.00 | 0.00 | 0.00 | 0.00 | 0.00 | 0.00 | 0.00 | 0.00 | 0.00 | 0.00 | 0.00 |
| ROME | 0.97 | 0.93 | 0.72 | 0.19 | 0.16 | 0.03 | 0.05 | 0.04 | 0.01 | 0.02 | 0.01 | 0.00 | 0.00 | 0.00 | 0.01 | 0.00 | 0.00 | 0.01 |
| MEMIT | 0.83 | 0.72 | 0.96 | 0.81 | 0.72 | 0.92 | 0.05 | 0.04 | 0.01 | 0.02 | 0.01 | 0.00 | 0.00 | 0.00 | 0.01 | 0.00 | 0.00 | 0.01 |
| AlphaEdit | **0.99** | 0.93 | **1.00** | **0.98** | **0.89** | 0.92 | **0.95** | **0.84** | **0.79** | 0.91 | **0.78** | 0.72 | 0.90 | **0.74** | 0.67 | 0.89 | **0.72** | 0.62 |
| NMKE (Ours) | 0.98 | **0.95** | **1.00** | 0.96 | 0.86 | **0.93** | **0.95** | **0.84** | **0.79** | **0.93** | 0.76 | **0.75** | **0.91** | 0.73 | **0.70** | **0.90** | 0.70 | **0.65** |

Table 8: Editing performance on the ZsRE dataset using Qwen2.5-7B across sequential steps.

| Method | $T=100$ | | $T=500$ | | $T=1000$ | | $T=2000$ | |
|---|---|---|---|---|---|---|---|---|
| | Rel. | Gen. | Rel. | Gen. | Rel. | Gen. | Rel. | Gen. |
| Alphaedit | **0.98** | **0.96** | **0.97** | 0.90 | 0.95 | 0.89 | 0.92 | 0.85 |
| NMKE | 0.97 | **0.96** | **0.97** | **0.93** | **0.97** | **0.91** | **0.96** | **0.90** |

## B.6 Hierarchical Distribution of Knowledge Neurons Across Layers

Figure 11 shows the distribution of knowledge-general and knowledge-specific neurons across layers 0–31 of LLaMA3-8B to reveal their structural roles in knowledge representation. Knowledge-general neurons are concentrated in the top layers, with a notably higher activation rate in layer 31, indicating that they play a central role in aggregating high-level knowledge. Knowledge-specific neurons are more evenly spread and gradually thin out with depth. The overlap ratio between knowledge-general and knowledge-specific neurons reaches its maximum in the middle layers, implying that these layers may act as integration hubs where general and specific knowledge intersect. This mirrors the theoretical intuition that intermediate Transformer layers blend low-level and high-level semantics.

Notably, instead of modifying all knowledge-associated neurons, our sparse mask dynamically identifies and edits only a minimal subset relevant to the target knowledge. The temperature coefficient and scaling factors were set from the empirical distribution of neuron activations rather than hand-tuned. All other hyperparameters were fixed across models and datasets, and NMKE consistently achieved strong editing and generalization.

## B.7 Memory overhead

To reduce overhead, we compute neuron attribution only in layers 4-8. As summarized in Table 9, each layer instantiates three additional memory objects: attribution coefficients ($\approx$0.33 MB per layer), a binary importance mask ($\approx$0.05 MB per layer), and two scalars. These objects are created on demand during forward and attribution and released after editing, and their memory cost is negligible relative to the model parameters.

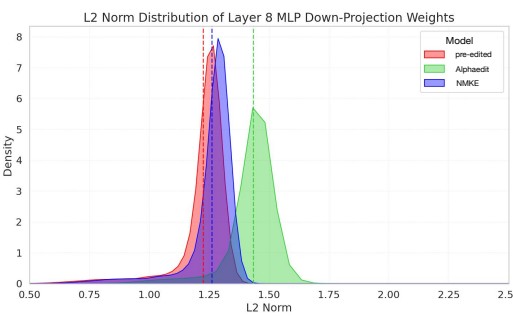

Figure 10: Hierarchical Distribution of Knowledge Neurons Across Layers

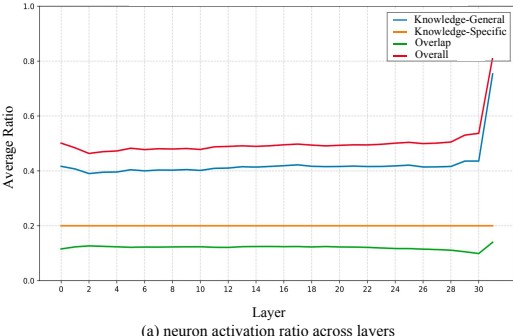

Figure 11: Hierarchical Distribution of Knowledge Neurons Across Layers

Table 9: Layer-wise attribution memory statistics.

| Layer | Coeff Size (MB) | Non-zero Neurons | Resonance Ratio | Burst Ratio |
|---|---|---|---|---|
| 4–8 | $0.33 \times 5$ | $\sim$6,072–7,320 | $\sim$0.30–0.33 | $\sim$0.24–0.28 |

Table 10: Quantitative analysis of parameter stability.

| Layer | Comparison | Wasserstein↓ | Cosine Mean↑ | Cosine Std↓ | $\ell_2$ Mean↓ | $\ell_2$ Std↓ |
|---|---|---|---|---|---|---|
| 7 | Base vs AlphaEdit | 0.2079 | 0.7632 | 0.0292 | 0.9337 | 0.1507 |
|   | Base vs NMKE | 0.0357 | 0.9588 | 0.0067 | 0.3564 | 0.0218 |
| 8 | Base vs AlphaEdit | 0.4371 | 0.6386 | 0.0594 | 1.2864 | 0.4421 |
|   | Base vs NMKE | 0.1057 | 0.8907 | 0.0175 | 0.6000 | 0.0542 |
| 9 | Base vs AlphaEdit | 0.0000 | 1.0000 | 0.0000 | 0.0000 | 0.0000 |
|   | Base vs NMKE | 0.0000 | 1.0000 | 0.0000 | 0.0000 | 0.0000 |

Table 11: Batched sequential editing scenarios.

| Method | $T = 1000$ | | | $T = 2000$ | | |
|---|---|---|---|---|---|---|
|  | Rel. | Gen. | Fluency | Rel. | Gen. | Fluency |
| AlphaEdit | 0.88 | 0.79 | 4.79 | 0.68 | 0.58 | 4.77 |
| NMKE | **0.92** | **0.83** | **5.82** | **0.90** | **0.80** | **5.79** |

## B.8 Quantitative analysis of parameter stability

We quantified weight shifts in layers 7–9 using Wasserstein, cosine, and $\ell_2$ distances. As shown in Table 10, NMKE induces significantly smaller distributional shifts than AlphaEdit in the edited layers (4–8), better preserving the internal weight structure. In the unedited layer 9, neither method produces any change.

## B.9 Batched sequential editing scenarios

Table 11 shows that we use a batch size of 4 across 2000 sequential edits to simulate practical simultaneous updates, evaluating all methods only after completing all edits for a fairer comparison. Our results demonstrate that NMKE preserves stable editing accuracy and locality as the number of edits grows, while AlphaEdit's performance degrades considerably. This is likely due to AlphaEdit's fixed global projection matrix, which cannot adapt to distributional shifts across multiple edits. In contrast, NMKE's fine-grained neuron-level masking minimizes per-batch weight changes, reducing model drift.

## C Limitations and Future Discussions

NMKE's performance largely depends on neuron attribution and dynamic sparse masking. Our current approach labels neurons as knowledge-general or knowledge-specific based on activation patterns. In practice, neuron responses vary along a continuum and are influenced by both within-layer activations and cross-layer interactions. Building on this, we will design more precise, continuous importance measures at both the intra-layer and inter-layer levels and explicitly model cross-layer information flow to learn path-aware attribution and masks that remain consistent across layers, which should improve parameter localization and stability during long-horizon editing. In addition, edits should satisfy dependency closure across related but distinct facts. For example, updating the current CEO should remain consistent with the total number of CEOs. Quantifying and mitigating post-edit propagation errors and conflicts is a key component of knowledge editing. These directions will guide our future work.

