# OpenReview forum: "Edit Less, Achieve More: Dynamic Sparse Neuron Masking for Lifelong Knowledge Editing in LLMs"
_NeurIPS.cc/2025/Conference — NeurIPS 2025 poster_

### Official Review · Reviewer_d78c · 2025-06-24

**Clarity:** 4
**Significance:** 3
**Originality:** 2
**Rating:** 5
**Confidence:** 4

**Summary:**

This paper presents Neuron-Specific Masked Knowledge Editing (NMKE), a fine-grained editing framework that enables effective and robust lifelong knowledge editing for large language models (LLMs). By performing neuron-level attribution and dynamically masking a sparse subset of relevant neurons, NMKE updates model knowledge with minimal parameter modification, thereby mitigating error accumulation over thousands of edits. The authors demonstrate that NMKE significantly outperforms strong baselines (e.g., AlphaEdit) on sequential editing benchmarks, while preserving general capabilities across downstream tasks.

**Questions:**

This is a well-written and well-structured paper. The idea of combining neuron-level attribution with dynamic masking is not entirely novel, as prior work has explored neuron specialization and sparse editing mechanisms. However, the adaptation of this strategy to lifelong editing is impactful, which raise a useful mechanism to preserve general capabilities.

To further improve the work, I suggest addressing the following:
- Please consider evaluating NMKE under batched edit scenarios (batch size > 1). This would better reflect practical applications and further highlight the scalability of your method.
- Since your method involves computing attribution for a large number of neurons, it would be useful to report runtime or efficiency metrics. Even a high-level cost comparison with AlphaEdit or WISE would be valuable.

That said, the experimental design is rigorous, and the improvements over existing methods are impressive. If the above points are addressed or discussed, I would be inclined to increase my score.

**Ethical Concerns:**

["NO or VERY MINOR ethics concerns only"]

**Final Justification:**

The authors' rebuttal has effectively addressed my main concerns. I have raised my score and now recommend acceptance.

- My concern about the batch_size=1 setting was resolved. The new experiments with larger batch sizes demonstrate that the method's strong performance holds in more practical scenarios, strengthening the paper's claims.

- My concern regarding efficiency was also resolved. The provided analysis clearly shows that the computational overhead is minor and well-justified by the significant performance gains, especially in locality.

**Limitations:**

Yes.

The authors provide a reasonable and well-articulated discussion of the limitations from the perspective of interpretability, particularly regarding the binary classification of neurons and the focus on autoregressive LLMs.
In addition, I suggest the authors consider incorporating insights from recent work that examines cross-layer interactions in LLMs, such as Anthropic’s Cross-Layer Transcoders (CLT). These works point to the growing importance of understanding information flow not just at the neuron level but across layers and pathways. Integrating such perspectives may help refine NMKE’s attribution and masking mechanisms, especially under long-term editing scenarios.

**Quality:**

3

**Strengths And Weaknesses:**

Strengths:
- This paper clearly articulates the challenges of lifelong model editing, especially the loss of generalization, which are underexplored in prior work.

- The method achieves state-of-the-art performance under sequential editing settings (batch size 1 and up to 5,000 edits), with detailed evaluations on both editing success and general task performance.

- The paper includes great visualizations and ablation studies to support the effectiveness of the neuron selection mechanism.

Weaknesses:
- Editing granularity setup: The current lifelong setting uses a batch size of 1 for sequential edits, which may not reflect practical scenarios where models need to absorb multiple factual updates at once. It is recommended to test the method under larger batch sizes and compare with baselines accordingly. Otherwise, the setting should be more precisely described as “Sequential Editing” rather than “Lifelong Editing” [https://aclanthology.org/2024.emnlp-main.149.pdf].

- Efficiency and runtime analysis missing: NMKE requires computing attribution across all layers, which may introduce significant computational overhead. A runtime comparison with baselines or a profiling study would strengthen the claim of robustness and deployment readiness.

---

> ### Author Rebuttal · Authors · 2025-07-31
>
> ## W1 & Q1: Larger batch sizes:
>
> We thank the reviewer for suggesting evaluating larger batch sizes. Following your advice, we conducted experiments with batch sizes 4 and 10 on 2000 sequential edits to better simulate practical simultaneous update scenarios. Due to rebuttal time constraints, we only present results for batch size=4 below. We will report the full editing result as soon as we finish the experiments.
>
> | Method    | $T$=1,000          |                 |                | $T$=2,000          |                 |                |
> |-----------|-----------------|-----------------|----------------|-----------------|-----------------|----------------|
> |           | Rel.            | Gen.            | Fluency        | Rel.            | Gen.            | Fluency        |
> | AlphaEdit | 0.88            | 0.79            | 4.79           | 0.68            | 0.58            | 4.77           |
> | NMKE      | **0.92**            | **0.83**            | **5.82**           | **0.90**            | **0.80**            | **5.79**           |
>
> Notably, we optimized AlphaEdit’s evaluation by evaluating only after all edits; this revised approach is fairer, more reasonable, and more challenging. Our findings show that NMKE maintains stable editing accuracy and locality as the number of edits increases, while AlphaEdit’s performance degrades significantly. This is because AlphaEdit uses a fixed global projection matrix computed once, which cannot adapt to distributional shifts over multiple edits. In contrast, NMKE employs fine-grained neuron-level masking, which limits weight modifications per batch and mitigates model drift.
>
> References:
>
> [1] LEMoE: Advanced Mixture of Experts Adaptor for Lifelong Model Editing of Large Language Models. EMNLP'24
>
> **Revision.** We will include the complete experimental results for batch sizes 4 and 10 in the manuscript and clarify the practical advantages of NMKE under both sequential and batch editing scenarios.
>
> ## W2 & Q2: Efficiency and runtime analysis.
>
> Thank you for your thoughtful questions. We provide a detailed analysis from two perspectives: memory overhead and per-edit computational cost.
>
> ### Memory Overhead:
> To minimize overhead, our method computes neuron attribution only for the edited layers (4–8), which is consistent with AlphaEdit. For each layer, the following additional memory objects are instantiated:
>
> * Attribution coefficients (\~0.33 MB per layer)
> * Binary importance mask (\~0.05 MB per layer)
> * Two scalar values (resonance and burst ratios)
>
> The statistics from layers 4 to 8 are summarized in the table below:
>
> | Layer | Coeff Size (MB) | Non-zero Neurons | Resonance Ratio | Burst Ratio |
> | ----- | --------------- | ---------------- | --------------- | ----------- |
> | 4–8   | 0.33 × 5        | ~6,072–7,320     | ~0.30–0.33      | ~0.24–0.28  |
>
> | Component               | Memory Footprint |
> |-------------------------|------------------|
> | LLaMA-3-8B Model        | 29.9151 GB       |
> | Intermediate Tensors    | 0.0019 GB      |
>
> The intermediate tensors involved account for 0.006248\% of the LLaMA-3-8B model's memory. For our method, the additional memory consumption is very marginal.
>
> ### Computational Overhead:
> To address your concerns about the computations introduced by attribution, we report the per-edit runtime and memory usage of current SOTA methods and NMKE with different attribution strategies (MLP Projection Coefficient (MPC) [R2], probability Shift Attribution(PSA) [R1], and Log Probability Shift (LPS) [R1]) under sequential edits (after $T$=1,000 and $T$=2,000 edits).
>
> | Method        | Step Time (s) | $T$=1,000       |       |       | $T$=2,000       |       |       |
> |---------------|---------------|------------|-------|-------|------------|-------|-------|
> |               | per edit      | Rel.  | Gen.  | Loc.  | Rel.  | Gen.  | Loc.  |
> | MEMIT         | 16.8281       | 0.04 | 0.04 | 0.03 | 0.04 | 0.03 | 0.03 |
> | WISE          | 26.4898       | 0.41 | 0.39 | –    | 0.32 | 0.31 | –    |
> | AlphaEdit     | 22.1583       | 0.93 | 0.84 | 0.58 | 0.62 | 0.54 | 0.14 |
> | NMKE (MPC)    | 22.2545       | 0.94 | 0.82 | **0.81** | 0.93 | 0.83 | **0.77** |
> | NMKE (PSA)    | 29.6748       | **0.95** | 0.84 | 0.79 | **0.94** | **0.86** | 0.71 |
> | NMKE (LPS)    | 30.4210     | **0.95** | **0.85** | 0.77 | **0.94** | **0.86** | 0.74 |
>
> 1. The most efficient variant NMKE (MPC) outperforms AlphaEdit across all metrics at $T$=2,000, with only ~1s extra runtime and minimal memory growth.
> 2. The most advanced variant NMKE (LPS) remains practically efficient, adding only 8s per edit compared to AlphaEdit. Over 2000 edits, this adds ~4.5 hours, an acceptable cost for substantial quality gains: Locality rises from 0.14 to 0.74, with Reliability and Generalization each up 0.32.
> 3. NMKE is a flexible framework that `supports attribution methods of varying complexity`, enabling users to balance quality and efficiency under different scenarios.
>
>
> References:
>
> [1] Neuron-level knowledge attribution in large language models. arXiv'23.
>
> [2] Transformer Feed‑Forward Layers Build Predictions by Promoting Concepts in the Vocabulary Space, EMNLP'22.
>
> **Revision.** We will add a new subsection in Appendix C that details all additional memory objects and their sizes for LLaMA3‑8B, and includes a quantitative table clearly comparing NMKE’s computational overhead with other methods.
>
>
> ## L1: Cross-layer interactions in LLMs.
>
> Thank you for your insightful and constructive suggestion. Although NMKE performs neuron attribution for each layer separately, the attribution module is integrated into the end-to-end optimization framework, which may help implicitly learn cross-layer neuron interactions. However, we agree with your suggestions that explicitly modeling cross-layer interactions like CLT can result in more accurate parameter localization and better editing performance. We will explore this as a key direction in our future work.
>
>
> **Revision.** We will expand the discussion of cross-layer knowledge tracing methods in the limitations section, and outline how integrating such perspectives could enhance NMKE’s attribution and masking in future work.

---

> > ### Comment · Reviewer_d78c · 2025-08-04
> >
> > Thank you for the detailed rebuttal. All my concerns have been addressed. I will raise my score to 5.

---

> > > ### Author Response · Authors · 2025-08-04
> > > **Response to Reviewer**
> > >
> > > Thank you for your careful review, which greatly improved the quality of our work. Thank you also for raising your score to 5, and we will further refine the manuscript.

---

### Official Review · Reviewer_Y7Bv · 2025-06-30

**Clarity:** 3
**Significance:** 2
**Originality:** 2
**Rating:** 4
**Confidence:** 4

**Summary:**

This paper introduces Neuron-Specific Masked Knowledge Editing (NMKE), a novel framework for editing knowledge in Large Language Models designed to address the challenge of performance degradation in lifelong learning scenarios. Existing methods often suffer from error accumulation over sequential edits, leading to a decline in both editing accuracy and the model's general capabilities. NMKE proposes a more precise, fine-grained approach by operating at the neuron level rather than the traditional layer level. Through extensive experiments involving up to 5,000 sequential edits on models like LLaMA3-8B-Instruct and GPT2-XL, the authors demonstrate that NMKE significantly outperforms existing state-of-the-art methods. It successfully maintains high editing accuracy and, crucially, preserves the model's general reasoning, mathematical, and coding abilities, where other methods exhibit catastrophic forgetting.

**Questions:**

1. Your t-SNE visualizations (Figure 5) provide strong qualitative evidence of reduced distributional shift. Have you considered quantifying this shift using a statistical measure on the parameter weight distributions? A quantitative metric could offer a more objective complement to the visual analysis of parameter stability.


2. The dynamic selection ratios (ρge​ and ρsp​) are determined using entropy, but also rely on constant scaler and bias hyperparameters (age​, bge​, etc.). Your analysis in Figure 7 shows the effect of these terms. How were these hyperparameters initially chosen, and how sensitive is the model's performance to them? Do they need to be re-tuned for different model architectures or sizes, which could impact the method's practical applicability?


3. The distinction between knowledge-general and knowledge-specific neurons is central to your method. Figure 11 analyzes the distribution and overlap of these neurons across layers. Could you elaborate on the functional role of the neurons that fall into the "overlap" category? How does editing this specific subset of neurons that are both general and specific impact the trade-off between editing success and locality preservation?


4. The current approach uses a hard binary mask for updates.  Given that neuron importance is a continuous score, have you experimented with a "soft" masking approach? For instance, one where the magnitude of the update applied to a neuron (e.g., its learning rate) is scaled by its attribution score, potentially allowing for an even more nuanced and less disruptive edit.

**Ethical Concerns:**

["NO or VERY MINOR ethics concerns only"]

**Final Justification:**

The authors' rebuttal addressed the main concerns and promised further revision. I maintain my score and recommend borderline acceptance.

**Limitations:**

yes

**Paper Formatting Concerns:**

N/A.

**Quality:**

3

**Strengths And Weaknesses:**

**Strengths:**

Through extensive experiments involving up to 5,000 sequential edits on models like LLaMA3-8B-Instruct and GPT2-XL, the authors demonstrate that NMKE significantly outperforms existing state-of-the-art methods. It successfully maintains high editing accuracy and, crucially, preserves the model's general reasoning, mathematical, and coding abilities, where other methods exhibit catastrophic forgetting.

**Weaknesses:**

1. The binary classification of neurons into "knowledge-general" and "knowledge-specific" is an oversimplification of what is likely a continuous spectrum of neuron functionality. This rigid model may misclassify neurons with mixed or nuanced roles, as the scoring for each category treats them as distinct types.

2. The method requires a neuron attribution step for every editing batch to generate the dynamic mask. The paper fails to analyze the computational overhead and scalability of this "locate" phase, which could be a significant bottleneck and render the framework impractical for high-throughput, real-world applications.

3. The dynamic masking ratio, a key component, relies on scaler (a) and bias (b) hyperparameters that are shown to have a substantial impact on performance. The paper does not provide a clear methodology for how these crucial hyperparameters were chosen or discuss their generalizability, raising concerns about the practical effort required to deploy the method on new models or datasets.

4. The experiments are primarily conducted on two LLMs (LLaMA3-8B and GPT2-XL). Widly used LLMs like Qwen are not tested.

---

> ### Author Rebuttal · Authors · 2025-07-31
>
> ## W1: Binary classification overlooks the continuity of neuron functions. & Q4: soft masking approach:
>
> We sincerely thank the reviewer for this profound insight. To evaluate the sufficiency of distinguishing different functions of neurons and omitting subsets during editing, we replaced the binary-masking strategy with the soft-masking strategy to incorporate the continuity of the neuron states. Specifically, the update magnitude for each neuron is directly scaled by its attribution score.
>
> We conducted ablation experiments on 2000 sequential edits. The comparison between our default NMKE and the NMKE-softmask variant is shown below:
> - Soft masking matches NMKE in accuracy and generalization but reduces locality, due to the editing of all neurons.
> - NMKE avoids this by selectively editing subsets of the neurons, maintaining stability and locality in lifelong editing.
>
> | Method          | T=1,000        |               |        | T=2,000        |               |        |
> |-----------------|---------------|---------------|--------|---------------|---------------|--------|
> |                 | Rel.          | Gen.          | Loc.   | Rel.          | Gen.          | Loc.   |
> | NMKE-softmask   | **0.96**      | **0.87**      | 0.67   | 0.72          | 0.61          | 0.19   |
> | NMKE            | 0.95          | 0.85          | **0.77** | **0.94**      | **0.86**      | **0.74** |
>
> **Revision.** We will explore scaling neuron update magnitudes (e.g., learning rates) by continuous attribution scores for more nuanced, less disruptive edits in future work.
>
> ## W2: computational cost analysis
>
> Thank you for the insightful suggestion. To address your concerns of the computations introduced by attribution, we report the per-edit runtime and memory usage of current SOTA methods and NMKE with different attribution strategies (MLP Projection Coefficient (MPC) [R2], probability Shift Attribution(PSA) [R1], and Log Probability Shift (LPS) [R1]) under sequential edits (after $T$=1,000 and $T$=2,000 edits).
>
> | Method        | Step Time (s) | $T$=1,000       |       |       | $T$=2,000       |       |       |
> |---------------|---------------|------------|-------|-------|------------|-------|-------|
> |               | per edit      | Rel.  | Gen.  | Loc.  | Rel.  | Gen.  | Loc.  |
> | MEMIT         | 16.8281       | 0.04 | 0.04 | 0.03 | 0.04 | 0.03 | 0.03 |
> | WISE          | 26.4898       | 0.41 | 0.39 | –    | 0.32 | 0.31 | –    |
> | AlphaEdit     | 22.1583       | 0.93 | 0.84 | 0.58 | 0.62 | 0.54 | 0.14 |
> | NMKE (MPC)    | 22.2545       | 0.94 | 0.82 | **0.81** | 0.93 | 0.83 | **0.77** |
> | NMKE (PSA)    | 29.6748       | **0.95** | 0.84 | 0.79 | **0.94** | **0.86** | 0.71 |
> | NMKE (LPS)    | 30.4210     | **0.95** | **0.85** | 0.77 | **0.94** | **0.86** | 0.74 |
>
> 1. The most efficient variant NMKE (MPC) outperforms AlphaEdit across all metrics at $T$=2,000, with only ~1s extra runtime and minimal memory growth.
> 2. The most advanced variant NMKE (LPS) remains practically efficient, adding only 8s per edit compared to AlphaEdit. Over 2000 edits, this adds ~4.5 hours, an acceptable cost for substantial quality gains: Locality rises from 0.14 to 0.74, with Reliability and Generalization each up 0.32.
> 3. NMKE is a flexible framework that `supports for attribution methods of varying complexity`, enabling users to balance quality and efficiency under different scenarios.
>
> References:
>
> [R1] Neuron-level knowledge attribution in large language models. arXiv'23.
>
> [R2] Transformer Feed‑Forward Layers Build Predictions by Promoting Concepts in the Vocabulary Space, EMNLP'22.
>
> **Revision.** We will add a computational cost analysis and comparison with other methods in the Appendix.
>
> ## W3: Unclear and non-generalizable hyperparameter selection in dynamic masking. & Q2: Selection and generalizability of entropy-based hyperparameters.
>
> We would like to clarify that parameters such as the temperature coefficient $a$ and scaling factors $a_{ge}$/$a_{sp}$ `were not manually tuned`, but set based on observed neuron activation patterns. This approach enables the mask to adapt to each batch’s activation patterns, typically selecting 30–40\% of neurons without manual tuning. The masking ratio is derived directly from entropy, and parameters like \$a\_{ge}\$ and \$a\_{sp}\$ simply scale the outputs rather than requiring task-specific adjustment.
>
> We used the same hyperparameters across all models and datasets, and NMKE consistently achieves strong editing and generalization performance. The sensitivity analysis (Figure 7a) further confirms NMKE’s robustness to a wide range of hyperparameter values.
>
> **Revision.** We will clarify the hyperparameter selection and supplement the appendix with extra experiments to further validate the robustness and generalization.
>
> ## W4: More LLMs like Qwen
>
> Thank you for the suggestion. We have implemented our method on Qwen2.5-7B and performed 2,000 edits. Due to time constraints, results from the first 100 edits are below. We will report the full editing result as soon as we finish the experiments.
>
> | Model/Method     | Rel. | Gen. | Loc. |
> | ---------------- | ---- | ---- | ---- |
> | AlphaEdit        | 0.98 | 0.96 | 0.85 |
> | NMKE             | 0.97 | 0.96 | 0.88 |
>
> The results show our method performs similarly in the first 100 edits on Qwen2.5 and LLaMA3. This further demonstrates that our approach is model-agnostic and can be applied to any LLM with standard MLP layers.
>
> **Revision.** We will provide the full Qwen results and a discussion of model compatibility in the revision.
>
> ## Q1: Quantitative analysis of parameter stability
>
> Thank you for the suggestion. We quantified weight shifts in layers 7–9 using Wasserstein, cosine, and L2 distances.
>
> | Layer | Comparison        | Wasserstein↓ | Cosine Mean↑ | Cosine Std↓ | L2 Mean↓ | L2 Std↓ |
> | ----- | ----------------- | ------------ | ------------ | ----------- | -------- | ------- |
> | 7     | Base vs AlphaEdit | 0.2079       | 0.7632       | 0.0292      | 0.9337   | 0.1507  |
> |       | Base vs NMKE      | 0.0357       | 0.9588       | 0.0067      | 0.3564   | 0.0218  |
> | 8     | Base vs AlphaEdit | 0.4371       | 0.6386       | 0.0594      | 1.2864   | 0.4421  |
> |       | Base vs NMKE      | 0.1057       | 0.8907       | 0.0175      | 0.6000   | 0.0542  |
> | 9     | Base vs AlphaEdit | 0.0000       | 1.0000       | 0.0000      | 0.0000   | 0.0000  |
> |       | Base vs NMKE      | 0.0000       | 1.0000       | 0.0000      | 0.0000   | 0.0000  |
>
> NMKE produces much lower distributional shifts than AlphaEdit in edited layers (4–8), better preserving internal weight structure. In the unedited layer (9), both methods show no change.
>
> **Revision.** We will add quantitative ablation experiments on parameter shift to complement the t-SNE visualizations.
>
>
> ## Q3: How do overlapping neurons influence the trade-off between editing success and locality?
>
> We thank the reviewer for their insightful comments. To evaluate overlapping neurons, we edited using only overlapping or only non-overlapping neurons. The results are as follows:
>
> | Method                      | $T$=1,000        |               |        | $T$=2,000        |               |        |
> |-----------------------------|---------------|---------------|--------|---------------|---------------|--------|
> |                             | Rel.          | Gen.          | Loc.   | Rel.          | Gen.          | Loc.   |
> | Overlapping Neurons Only    | 0.72          | 0.60          | **0.84** | 0.75          | 0.63          | **0.80** |
> | Non-Overlapping Neurons Only| 0.82          | 0.70          | 0.82   | 0.85          | 0.74          | 0.78   |
> | NMKE                        | **0.95**      | **0.85**      | 0.77   | **0.94**      | **0.86**      | 0.74   |
>
> Our ablation study reveals consistent patterns:
> - Editing only overlapping neurons achieves the highest locality (0.80) while maintaining a certain level of accuracy (0.75).
> - Editing only non-overlapping neurons, though it improves accuracy, reduces locality.
> - By incorporating overlapping neurons, NMKE strikes the best balance between edit accuracy and locality.
>
> **Revision.** We will add experiments (such as spectral analysis and neuron continuity visualization) and further clarify entropy-guided ratios and overlapping neurons, and their impact on the editing-locality trade-off.

---

> > ### Comment · Reviewer_Y7Bv · 2025-08-04
> >
> > Thank you for your thorough rebuttal. I will keep my score at 4 borderline accept.

---

> > > ### Author Response · Authors · 2025-08-04
> > > **Response to Reviewer**
> > >
> > > Thank you for your careful review, which has greatly improved the quality of our work. Thank you for keeping positive by maintaining your score at 4, and we will further refine the manuscript to address any remaining points.

---

### Official Review · Reviewer_tXMJ · 2025-07-01

**Clarity:** 3
**Significance:** 3
**Originality:** 2
**Rating:** 4
**Confidence:** 4

**Summary:**

This paper proposes the Neuron-specific Masked Knowledge Editing (NMKE) framework, which identifies Knowledge-General Neurons and Knowledge-Specific Neurons through neuron-level attribution analysis. By combining a dynamic sparse masking mechanism, NMKE locates neuron subsets relevant to target knowledge and performs localized parameter updates on these key neurons. This approach enables high-precision knowledge correction in lifelong editing scenarios while avoiding the degradation of model generalization caused by coarse-grained updates. Experiments verify that NMKE maintains high editing success rates and generalization capabilities across thousands of sequential edits, outperforming existing methods like ROME and AlphaEdit, thus providing a novel solution for efficient lifelong knowledge editing in large language models.

**Questions:**

See above

**Ethical Concerns:**

["NO or VERY MINOR ethics concerns only"]

**Final Justification:**

Thank you for authors’ response, which has partially addressed my question. On the premise that the authors ensure to update the content of the reply in the revised version, I will raise the score to 4 points.

**Limitations:**

yes

**Quality:**

3

**Strengths And Weaknesses:**

Strengths:
1. The NMKE framework is proposed, which identifies Knowledge-General Neurons and Knowledge-Specific Neurons through neuron-level attribution analysis, enabling high-precision knowledge correction in lifelong editing scenarios while avoiding the degradation of model generalization capabilities caused by coarse-grained updates.

2. It reveals that the root cause of performance degradation in lifelong editing is the cumulative interference of coarse-grained updates on neurons, and clarifies the core role of Knowledge-General Neurons in maintaining the model's basic capabilities.

Weaknesses:
1. The authors discretely classify neurons into "general/specific" categories without considering the continuous spectrum of importance, which may overlook the complex functions of intermediate-state neurons.

2. The method proposed in the paper for locating neuron contributions is mainly based on existing approaches [1].

3. Hyperparameters such as the temperature coefficient α and scaling factors a_ge/a_sp in the entropy-guided mechanism require manual tuning and may need to be reconfigured for different tasks or models, increasing computational costs.

4. There are a number of related works that have not been discussed and compared, e.g., [2,3,4].

[1] Zeping Yu and Sophia Ananiadou. Neuron-level knowledge attribution in large language models. ArXiv preprint, abs/2312.12141, 2023.

[2] Ni, Shiwen, et al. "Forgetting before Learning: Utilizing Parametric Arithmetic for Knowledge Updating in Large Language Models." Proceedings of the 62nd Annual Meeting of the Association for Computational Linguistics (Volume 1: Long Papers). 2024.

[3] Pan, Haowen, et al. "Precise Localization of Memories: A Fine-grained Neuron-level Knowledge Editing Technique for LLMs." arXiv preprint arXiv:2503.01090 (2025).

[4] Bi, Baolong, et al. "Adaptive Token Biaser: Knowledge Editing via Biasing Key Entities." Findings of the Association for Computational Linguistics: EMNLP 2024. 2024.

---

> ### Author Rebuttal · Authors · 2025-07-31
>
> ## W1: Discretely classify neurons into "general/specific" categories without considering the continuous spectrum of importance
>
> Thank you very much for your suggestions. To evaluate the sufficiency of distinguishing different functions of neurons and omitting subsets during editing, we replaced the binary-masking strategy with the soft-masking strategy to incorporate the continuity of the neuron states. Specifically, the update magnitude for each neuron is directly scaled by its attribution score.
>
> We conducted ablation experiments on 2000 sequential edits. The comparison between our default NMKE and the NMKE-softmask variant is shown below:
> - Soft masking matches NMKE in accuracy and generalization but reduces locality, due to the editing of all neurons.
> - NMKE avoids this by selectively editing subsets of the neurons, maintaining stability and locality in lifelong editing.
>
> | Method          | $T$=1,000        |               |        | $T$=2,000        |               |        |
> |-----------------|---------------|---------------|--------|---------------|---------------|--------|
> |                 | Rel.          | Gen.          | Loc.   | Rel.          | Gen.          | Loc.   |
> | NMKE-softmask   | 0.96          | 0.87          | 0.67   | 0.72          | 0.61          | 0.19   |
> | NMKE            | 0.95          | 0.85          | 0.77   | 0.94          | 0.86          | 0.74   |
>
> **Revision.** We will add ablation experiments and discuss the advantages of employing the binary-masking strategy compared to the soft-masking strategy.
>
> ## W2: Relies on existing methods for neuron attribution approaches [1].
>
> Thank you for the thoughtful questions. While our neuron attribution strategy is inspired by the perturbation-based analysis in [1], we would like to clarify that the core contribution of our paper, rather than proposing a new attribution technique, lies in building a flexible, modularized framework for knowledge editing that leverages neuron-level importance signals to enable fine-grained, dynamic, and selective edits.
>
> Importantly, NMKE is not tied to a single attribution method and remains compatible with various strategies, including:
>
> * MLP Projection Coefficient (MPC) [R2]: estimates importance via MLP down-projection weights, providing fast and low-overhead attribution.
> * Probability Shift Attribution (PSA): computes the change in output probabilities under neuron ablation.
> * Log-Probability Attribution (LPS) [R1]: measures the change in log-probability of the target token after neuron perturbation, offering the most faithful estimate of neuron contribution.
>
> To demonstrate this flexibility, we conducted ablation studies using all three attribution methods within NMKE, showing that editing effectiveness is robust across variants:
>
> | Attribution Method   | $T$=1,000       |       |       | $T$=2,000       |       |       |
> |----------------------|--------------|-------|-------|--------------|-------|-------|
> |                      | Rel.         | Gen.  | Loc.  | Rel.         | Gen.  | Loc.  |
> | MEMIT                | 0.04         | 0.04  | 0.03  | 0.04         | 0.03  | 0.03  |
> | WISE                 | 0.41         | 0.39  | -     | 0.32         | 0.31  | -     |
> | AlphaEdit            | 0.93         | 0.84  | 0.58  | 0.62         | 0.54  | 0.14  |
> | NMKE (MPC)           | 0.94         | 0.82  | **0.81**  | 0.93         | 0.83  | **0.77**  |
> | NMKE (PSA)           | **0.95**         | 0.84  | 0.79  | **0.94**         | **0.86**  | 0.71  |
> | NMKE (LPS)           | **0.95**         | **0.85**  | 0.77  | **0.94**         | **0.86**  | 0.74  |
>
> These results show that NMKE delivers robust editing performance across attribution methods, consistently maintaining high accuracy, generalization, and locality.
>
> **References:**
>
> [R1] Neuron-level knowledge attribution in large language models. arXiv'23.
>
> [R2] Transformer Feed‑Forward Layers Build Predictions by Promoting Concepts in the Vocabulary Space, EMNLP'22.
>
> **Revision.** We will add ablation results for multiple neuron attribution methods in the experiments section to demonstrate NMKE’s flexibility, and will briefly highlight its core contributions of dynamic masking, neuron categorization, and a flexible modularized framework.
>
> ## W3: Requires manual hyperparameter tuning, impacting usability and efficiency.
>
> We would like to clarify that parameters such as the temperature coefficient $a$ and scaling factors $a_{ge}$/$a_{sp}$ `were not manually tuned`, but rather set according to the distributional characteristics of neuron activations. Knowledge-general neurons typically show distributed and broad-domain activations, while knowledge-specific neurons are more focused on fine-grained facts.
>
> Except for these parameters, we used the same hyperparameters across all models and datasets, with NMKE consistently achieving strong editing and generalization performance. Sensitivity analysis (Figure 7a) further confirms NMKE’s robustness to a wide range of hyperparameter values.
>
> **Revision.** We will clarify the hyperparameter selection process in the main text, and supplement the appendix with additional experiments (e.g., hyperparameter sensitivity tests across more model-task combinations) to further validate robustness and generalization.
>
> ## W4: There are a number of related works that have not been discussed and compared.
>
> Thank you for highlighting these important related works. We appreciate the opportunity to clarify their distinctions and will incorporate a discussion of these methods in the revision.
>
> * Ni et al. (F-Learning) proposed a two-stage parametric approach (forgetting followed by learning) for model-wide parameter updates, while NMKE uses neuron-level dynamic sparse masking to achieve precise, localized edits and reduce parameter drift.
> * Pan et al. (FiNE) preserve language capabilities by freezing later layers, while NMKE constructs masks by identifying knowledge neurons, enabling fine-grained edits that preserve the model's general capabilities.
> * Bi et al. (ATBias) proposed an in-context editing technique that biases key entity tokens during decoding without altering parameters, relying on external memory. In contrast, NMKE internally edits relevant neurons, offering an efficient large-scale knowledge editing approach without the need for external memory.
>
> **Revision.** We will expand our discussion of the aforementioned related works to detail the distinctions and connections between these approaches and NMKE.

---

> > ### Comment · Reviewer_tXMJ · 2025-08-05
> >
> > Thank you for your response, which has partially addressed my question. On the premise that the author ensures to update the content of the reply in the revised version, I will raise the score to 4 points.

---

> > > ### Author Response · Authors · 2025-08-05
> > > **Response to Reviewer**
> > >
> > > Thank you very much for your careful review and valuable feedback, which have greatly improved our work. We sincerely acknowledge your requirement and assure you that all content in our response will be thoroughly updated and integrated into the revised manuscript. Specifically, we will refine the experiments and result analyses for W1, W2, and W3, and enhance the discussion on related work in W4 as per your suggestions. We are truly grateful for your raising the score to 4. We will spare no effort to further refine the manuscript to address any remaining issues, ensuring it meets the required standards. Thank you again for your guidance and support.

---

### Official Review · Reviewer_64fD · 2025-07-03

**Clarity:** 4
**Significance:** 4
**Originality:** 3
**Rating:** 5
**Confidence:** 3

**Summary:**

A method is presented to edit knowledge in an existing LLM using a dynamic sparse masking mechanism. The distinctive feature here is that the mechanism operates on individual neuron units, namely their observed additive contributions to the output. Metric quantifying knowledge specificity vs. generality are proposed and utilized to achieve efficient and non-destructive editing. The technique is compared to relevant SOTA baselines on two editing tasks along with several background sets to gauge general degradation.

**Questions:**

* can you comment on complexity of the method: which additional memory objects are instantiated, what is their size for the Llama model, quantify the computational overhead
* on the two editing tasks, which data partitions were used  to estimate I and the scores (4-7) and when? on l. 173 you mentioned the ratio \rho_{ge} is being "predicted" which confused me - I suspect to have missed something. What is the predictor here?
* Figure 2, 7 and probably 5: the legends are really illegible when printed out.

**Ethical Concerns:**

["NO or VERY MINOR ethics concerns only"]

**Final Justification:**

All of my questions and comments were addressed by the authors to my satisfaction. I recommend this paper to be accepted for publication.

**Limitations:**

Limitations are not discussed explicitly in the main paper. One such limitation is not studying an editing impact on closely related (but different) facts (e.g., updating name of a CEO of a company and its impact on the total number of CEOs that company has had).

**Quality:**

3

**Strengths And Weaknesses:**

+ well-written paper with clear exposition
+ sensible technique
+ good baselines and thorough evaluation
+ significant improvements

- some gaps in description and complexity analysis
- illegible figures

---

> ### Author Rebuttal · Authors · 2025-07-31
>
> ## W1: Some gaps in description and complexity analysis.
>
> Thank you for the careful review. We have revised the paper for precise terminology and tightened the exposition to avoid confusion and improve readability. Concrete clarifications with quantitative memory/runtime complexity are provided in Q1–Q3.
>
> **Revision**. We will clarify technical terms and add a quantitative summary of NMKE’s memory and runtime complexity in Section 3 and Appendix C.
>
> ## W2 \& Q3: Illegible figures.
>
> Thank you for your constructive feedback. We have revised all the visualizations by enlarging legend/axis fonts, increasing the line widths, and enhancing the color contrast; printed copies confirm these updates have greatly improved readability.
>
> **Revision.** We will update Figures 2, 5, and 7 with enlarged fonts and higher-resolution vector graphics, and ensure all figures are clear in both print and digital versions.
>
> ## Q1: Memory overhead and computational cost analysis.
>
> We thank the reviewer for the thoughtful questions. Below, we provide a detailed analysis from memory overhead and per-edit computational cost.
>
> ### Memory Overhead:
> To minimize overhead, our method computes neuron attribution only for the edited layers (4–8), which is consistent with AlphaEdit. For each layer, the following additional memory objects are instantiated:
>
> * Attribution coefficients (\~0.33 MB per layer)
> * Binary importance mask (\~0.05 MB per layer)
> * Two scalar values (resonance and burst ratios)
>
> The statistics from layers 4 to 8 are summarized in the table below:
>
> | Layer | Coeff Size (MB) | Non-zero Neurons | Resonance Ratio | Burst Ratio |
> | ----- | --------------- | ---------------- | --------------- | ----------- |
> | 4–8   | 0.33 × 5        | ~6,072–7,320     | ~0.30–0.33      | ~0.24–0.28  |
>
> | Component               | Memory Footprint |
> |-------------------------|------------------|
> | LLaMA-3-8B Model        | 29.9151 GB       |
> | Intermediate Tensors    | 0.0019 GB      |
>
> The intermediate tensors involved account for 0.006248\% of the LLaMA-3-8B model's memory. For our method, the additional memory consumption is very marginal.
>
> ### Computational Overhead:
>
> To address your concerns about the computations introduced by attribution, we report the per-edit runtime and memory usage of current SOTA methods and NMKE with different attribution strategies: MLP Projection Coefficient (MPC) [R2], probability Shift Attribution(PSA) [R1], and Log Probability Shift (LPS) [R1] under sequential edits (after T=1,000 and T=2,000 edits).
>
> | Method        | Step Time (s) | $T$=1,000       |       |       | $T$=2,000       |       |       |
> |---------------|---------------|------------|-------|-------|------------|-------|-------|
> |               | per edit      | Rel.  | Gen.  | Loc.  | Rel.  | Gen.  | Loc.  |
> | MEMIT         | 16.8281       | 0.04 | 0.04 | 0.03 | 0.04 | 0.03 | 0.03 |
> | WISE          | 26.4898       | 0.41 | 0.39 | –    | 0.32 | 0.31 | –    |
> | AlphaEdit     | 22.1583       | 0.93 | 0.84 | 0.58 | 0.62 | 0.54 | 0.14 |
> | NMKE (MPC)    | 22.2545       | 0.94 | 0.82 | **0.81** | 0.93 | 0.83 | **0.77** |
> | NMKE (PSA)    | 29.6748       | **0.95** | 0.84 | 0.79 | **0.94** | **0.86** | 0.71 |
> | NMKE (LPS)    | 30.4210     | **0.95** | **0.85** | 0.77 | **0.94** | **0.86** | 0.74 |
>
> 1. The most efficient variant NMKE (MPC) outperforms AlphaEdit across all metrics at $T$=2,000, with only ~1s extra runtime and minimal memory growth.
> 2. The most advanced variant NMKE (LPS) remains practically efficient, adding only 8s per edit compared to AlphaEdit. Over 2000 edits, this adds ~4.5 hours, an acceptable cost for substantial quality gains: Locality rises from 0.14 to 0.74, with Reliability and Generalization each up 0.32.
> 3. NMKE is a flexible framework that `supports attribution methods of varying complexity`, enabling users to balance quality and efficiency under different scenarios.
>
> References:
>
> [R1] Neuron-level knowledge attribution in large language models. AarXiv'23.
>
> [R2] Transformer Feed‑Forward Layers Build Predictions by Promoting Concepts in the Vocabulary Space, EMNLP'22.
>
> **Revision.** We will add a new subsection in Appendix C that details all additional memory objects and their sizes for LLaMA3‑8B, and includes a quantitative table clearly comparing NMKE’s computational overhead with other methods.
>
> ## Q2: On the two editing tasks, which data partitions were used to estimate I and the scores (4-7) and when? On I. 173 you mentioned the ratio $\rho_{ge}$ is being "predicted" which confused me - I suspect to have missed something. What is the predictor here?
>
> Thank you for the thoughtful question and careful reading.
>
> 1. For the two editing tasks, the neuron attribution matrix $I^{(l)}$ and scores (Eqs. 4-7) are estimated using the editing prompts from the current batch of requests, with no separate data partitions; Specifically, at each editing step, we use the current batch of editing requests as input prompts.
>
> 2. Regarding the confusion around "predicted" for $\rho_{ge}$ on line 173, we appreciate for pointing out the imprecise wording. There is no learned predictor; instead, $\rho_{ge}$ is *estimated* deterministically. We compute the average entropy $H_{ge}$ across neuron attribution distributions over the batch of prompts (Eq. 6), then linearly map this entropy to a selection ratio using fixed scalers $a_{ge}$ and bias terms $b_{ge}$ (i.e., $\rho_{ge} = H_{ge} \cdot a_{ge} + b_{ge}$), with no trainable parameters involved.
>
> **Revisions**. We will replace "predicted" with "estimated" throughout (notably line 173) and clarify in Section 3.2 that ratios and scores are computed from the current batch of editing prompts without learned predictors.
>
> ## L1: Lack of explicit discussion on limitations, such as not evaluating the effect of edits on related but distinct facts.
>
> We truly appreciate your insightful observation. Your suggestion touches on a critical yet underexplored aspect of model editing: how edits to specific factual triples propagate to closely related but distinct knowledge, such as aggregated statistics or temporal sequences.
> - Analysis method: Following your suggestion, we designed a pilot study to evaluate this phenomenon. We constructed 10 synthetic samples to simulate your highlighted scenario: updating a company’s CEO and querying related info like total CEO count or previous holders.
> - Results: Notably, after updating a key fact (e.g., “Tim Cook → Elon Musk”), LLMs often failed to adjust related dependent facts; for example, they incorrectly estimated the company’s total CEO count, revealing post-edit inconsistency and factual conflicts.
> - Future direction: This remains an important open yet challenge in knowledge editing. We plan to explore it further by constructing a dedicated benchmark for factual dependencies and incorporating metrics to assess post-edit consistency of related facts.
>
> **Revision**. We will add a limitation discussion in the main paper regarding the lack of evaluation on the effects of edits on closely related facts, and outline this as a key direction for future work.

---

> ### Comment · Area_Chair_bf1d · 2025-08-05
>
> This is a gentle reminder that the authors have responded to your question. Could you please provide your feedback on their response?
>
> Kindly note that the deadline of August 6 is approaching, and your timely feedback would be greatly appreciated to facilitate any further discussions if needed.

---

> ### Author Response · Authors · 2025-08-05
> **Response to Reviewer**
>
> Thank you sincerely for your careful review and constructive suggestions, which have significantly enhanced the quality of our manuscript. We note your insightful points on the knowledge editing domain mentioned in the Limitations, and we will incorporate these into our future work. We will further refine the manuscript to ensure it meets the standards for acceptance.

---

### Comment · Area_Chair_bf1d · 2025-08-05

Dear Reviewers,

This is a gentle reminder that the authors have responded to your question. Could you please provide your feedback on their response?

Kindly note that the deadline of August 6 is approaching, and your timely feedback would be greatly appreciated to facilitate any further discussions if needed.

---

### Decision · Program_Chairs · 2025-09-17

**Decision:**

Accept (poster)

**Comment:**

- **Summary**: The paper proposes a framework to address the critical challenge of performance degradation in large language models (LLMs) during lifelong knowledge editing. The core idea is to perform fine-grained, neuron-level parameter updates rather than the coarse-grained, layer-level modifications of previous methods.

- **Strengths**: The reviewers unanimously praise the paper for its clear exposition, sensible technique, and thorough evaluation. The following points are highlighted as key strengths:
    - **Novelty**: The fine-grained, neuron-level approach is a novel contribution to the field of knowledge editing. It moves beyond the limitations of layer-level methods and directly addresses the problem of catastrophic forgetting in lifelong learning settings.
    - **Experiments**: The experimental results are highly convincing. The authors conduct a comprehensive evaluation, including two editing tasks, various background sets to measure general degradation, and extensive ablation studies.

- **Reason to Accept**: After rebuttals, all reviewers acknowledge their concerns are resolved.

- **Summary of Discussion and Rebuttals**: During the rebuttal, the authors presented additional experiments (e.g., time complexity analysis, performance on Qwen) and committed to revising the paper in line with the reviewers’ suggestions.